# LLEMA: Evolutionary Search with LLMs for Multi-Objective Materials Discovery

**Nikhil Abhyankar**[1*]  **Sanchit Kabra**[1*]  **Saaketh Desai**[2]  **Chandan K. Reddy**[1]
[1]Department of Computer Science, Virginia Tech
[2]Center of Integrated Nanotechnologies, Sandia National Laboratories

## Abstract

Materials discovery requires navigating vast chemical and structural spaces while satisfying multiple, often conflicting, objectives. We present **LLM**-guided **E**volution for **MA**terials discovery (LLEMA), a unified framework that couples the scientific knowledge embedded in large language models with chemistry-informed evolutionary rules and memory-based refinement. At each iteration, an LLM proposes crystallographically specified candidates under explicit property constraints; a surrogate-augmented oracle estimates physicochemical properties; and a multi-objective scorer updates success/failure memories to guide subsequent generations. Evaluated on **14** realistic tasks that span electronics, energy, coatings, optics, and aerospace, LLEMA discovers candidates that are chemically plausible, thermodynamically stable, and property-aligned, achieving higher hit rates and improved Pareto front quality relative to generative and LLM-only baselines. Ablation studies confirm the importance of rule-guided generation, memory-based refinement, and surrogate prediction. By enforcing synthesizability and multi-objective trade-offs, LLEMA provides a principled approach to accelerating practical materials discovery.

Code ⌂: `https://github.com/scientific-discovery/LLEMA`

Dataset 🤗: `https://huggingface.co/datasets/nikhilsa/LLEMABench`

## 1 Introduction

Materials discovery requires identifying or designing materials with properties tailored to a specific task. However, the immense combinatorial space of chemical and structural compositions makes the traditional discovery process resource-intensive and slow (Hautier et al., 2012; Davies et al., 2016). While machine learning has significantly accelerated materials screening, its effectiveness is often constrained by the availability of large, high-quality labeled datasets, limiting performance in data-scarce regimes (Chang et al., 2022; Xu et al., 2023). Trained on vast text corpora, including scientific literature, large language models (LLMs) offer a means to inject prior knowledge, making them promising tools for scientific discovery in data-scarce settings (White, 2023). Recently, LLMs have been leveraged to bridge natural language and material science, using textual knowledge to generate and refine hypotheses (Jia et al., 2024; Sprueill et al., 2024; Ghafarollahi & Buehler, 2025; Kumbhar et al., 2025). However, most existing methods rely on prompt engineering or unguided material generation, often producing candidates that are theoretically plausible yet unstable or impractical to synthesize. Moreover, they typically formulate materials discovery as a single-objective task, optimizing for one property (e.g., bandgap, stability, or conductivity) in isolation, whereas **real-world materials discovery is inherently multi-objective** (Gopakumar et al., 2018), requiring trade-offs among competing targets such as electrical conductivity and thermal resistance in thermoelectric materials (Hao et al., 2019).

To address these challenges, we propose **LLM**-guided **E**volution for **MA**terial discovery (**LLEMA**), *a novel agentic framework that integrates LLM scientific priors with evolutionary search and chemistry-informed design principles to autonomously generate, evaluate, and refine candidate materials under multiple task-specific property constraints.* LLEMA **introduces a set of chemistry-informed design**

---

*Equal contribution. Correspondence: `nikhilsa@vt.edu`, `sanchit23@vt.edu`.

**principles that act as operators to guide the LLM to generate and refine candidates iteratively**. These principles encode core knowledge across the entire materials discovery cycle, spanning compositional substitution, crystal structure manipulation, phase stability, and property-specific conditions. Unlike prior baselines, this chemically grounded generation integrates thermodynamic stability, enabling the systematic discovery of compounds that are both novel and experimentally realizable. At each iteration, the LLM fuses pretrained chemical knowledge with domain rules to balance exploration of chemical space and exploitation of promising leads. The candidates are then expressed as crystallographic information files (CIFs) for downstream property prediction (Figure 1B). Surrogate-assisted oracle models then estimate task-relevant physicochemical properties (Figure 1C), and candidates are scored against both performance objectives and design constraints. Successful and failed trajectories are fed back to the LLM (Figure 1D), enabling evolutionary refinement of subsequent generations. Thus, LLEMA provides a principled framework for multi-objective discovery by explicitly enforcing stability and synthesizability to generate compounds that are not only novel but also physically realizable.

Table 1 compares LLEMA with traditional and LLM-based materials discovery methods. Traditional generative methods require task-specific retraining that limits their generalizability across materials design problems. Furthermore, they lack the extensive prior knowledge embedded in LLMs and

**Table 1:** Existing materials discovery frameworks.

| Method | Domain Knowledge | Multi-objective Optimization | Rule-Guided Generation | Evolutionary Refinement |
|---|---|---|---|---|
| CDVAE (Xie et al., 2022) | ✗ | ✗ | ✗ | ✗ |
| G-SchNet (Gebauer et al., 2019) | ✗ | ✗ | ✗ | ✗ |
| DiffCSP (Jiao et al., 2023) | ✗ | ✗ | ✗ | ✗ |
| MatterGen (Zeni et al., 2025) | ✗ | ✗ | ✗ | ✗ |
| LLMatDesign (Jia et al., 2024) | ✓ | ✗ | ✗ | ✗ |
| **LLEMA (Ours)** | ✓ | ✓ | ✓ | ✓ |

feedback-driven refinement mechanisms. Methods like LLMatDesign employ sequential, single-step modifications with self-reflection to optimize candidates for a single property. In contrast, LLEMA combines chemistry-informed evolutionary operators, surrogate-augmented oracle, multi-objective scoring mechanisms, and memory-based evolution to achieve feedback-driven materials design.

To evaluate LLEMA, **we introduce a new benchmark suite of 14 diverse, industrially relevant discovery tasks**. The benchmark focuses on application-critical settings, covering problems relevant to electronics, energy, coatings, optics, and aerospace (Appendix C). Unlike existing benchmarks that focus on optimizing a single material property, each task in our suite is formulated as an explicit *multi-objective discovery* problem satisfying multiple properties. For example, discovering wide-bandgap semiconductors, a task critical for power and optoelectronic industries, requires simultaneously optimizing band gap and formation energy, rather than optimizing for a property in isolation. By capturing the complexity of real-world materials design, this benchmark provides a rigorous testbed for data-efficient, multi-objective discovery under physical and chemical constraints. Combining chemistry-informed design with iterative LLM-guided evolution, LLEMA goes beyond proposing candidates that are good at a single metric, instead generating *plausible, synthesizable materials* that satisfy complex, real-world objectives. We evaluated LLEMA using `GPT-4o-mini` (OpenAI, 2023) and `Mistral-Small-3.2-24B-Instruct-2506` (Mistral AI, 2025) as LLM backbones. Our results demonstrate that LLEMA consistently discovers chemically valid and structurally accurate candidates, with faster convergence across test settings. Our analysis further highlights the crucial role of rule-guided generation, memory-based feedback, and surrogate-assisted property prediction in LLEMA's performance. The contributions of this work are summarized as:

- **A synthesizability-aware evolutionary framework.** We introduce LLEMA, a framework that integrates LLMs' scientific knowledge with chemistry-informed evolutionary operators, explicitly enforcing chemical validity and thermodynamic feasibility during the search process.

- **Memory-based evolution.** We design a mechanism that leverages success and failure pools, together with multi-island sampling, to iteratively steer LLMs toward high-performing regions while avoiding memorization.

- **Constrained multi-objective formulation for materials discovery.** We cast materials design as a constrained multi-objective optimization problem, jointly optimizing competing property objectives.

- **Large-scale evaluation on realistic discovery tasks.** We construct a benchmark suite of 14 industrially motivated materials discovery tasks and demonstrate that LLEMA consistently achieves higher validity, stability, and Pareto efficiency than state-of-the-art baselines.

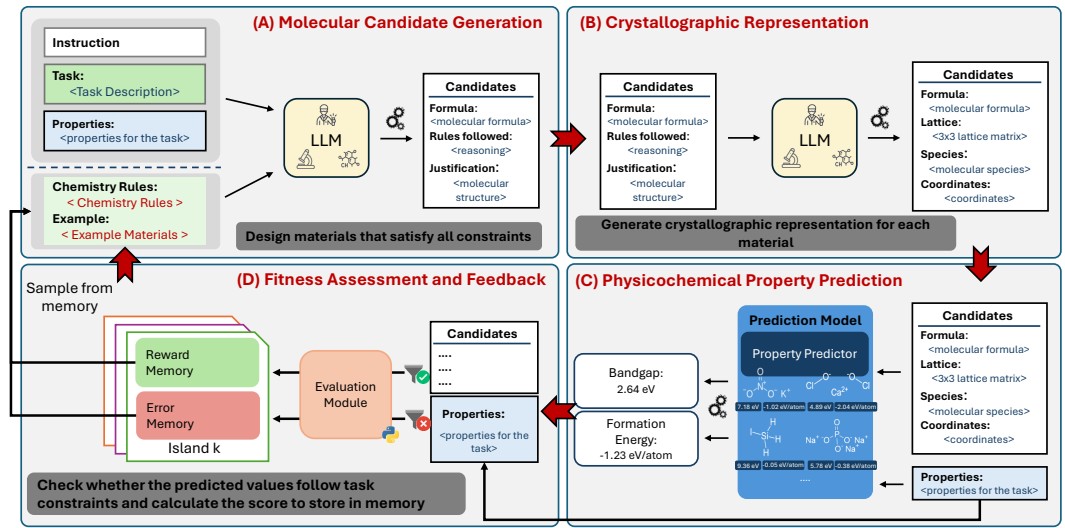

**Figure 1: LLEMA Framework,** consisting of four main components: (A) **Material Candidate Generation**, where an LLM generates candidates based on task descriptions and property constraints; (B) **Crystallographic Representation**, which converts generated materials into structured crystallographic information files (CIFs); (C) **Physicochemical Property Prediction**, to predict material properties such as formation energy and band gap, etc; and (D) **Fitness Assessment and Feedback**, which evaluates constraint satisfaction and provides iterative feedback through success/failure memory pools.

## 2 LLEMA METHOD

### 2.1 PROBLEM FORMULATION

We formulate the materials discovery task $\mathcal{T}$ as an optimization problem over the chemical space, where the goal is to identify the optimal material:

$$m^* = \arg\max_{m \in \mathcal{M}} f(m), \tag{1}$$

where $m$ denotes a material from the valid candidate set $\mathcal{M}$ representing the chemical space, and the function $f : \mathcal{M} \to \mathbb{R}$ is a black-box objective that assigns each material a scalar value to the property of interest. However, in practice, materials discovery rarely involves optimizing a single property. Instead, materials must satisfy multiple property constraints $\mathcal{C} = \{c_1, c_2, \ldots, c_k\}$ while jointly optimizing competing objectives $f_1, \ldots, f_n$. Each constraint $c_i$ corresponds to a property function $f_i : \mathcal{M} \to \mathbb{R}$ and takes one of the following canonical forms: an interval constraint, enforcing that the property lies within a feasible range; a lower-bound constraint, enforcing a minimum acceptable value; or an upper-bound constraint, enforcing a maximum allowable value. Formally,

$$c_i : f_i(m) \in [l_i, u_i] \quad \text{or} \quad c_i : f_i(m) \geq l_i \quad \text{or} \quad c_i : f_i(m) \leq u_i, \tag{2}$$

where $l_i, u_i \in \mathbb{R}$ denote constraint-specific thresholds. The overall task thus reduces to identifying candidates $m \in \mathcal{M}$ that satisfy all constraints while achieving favorable trade-offs across objectives. A naive strategy for multi-objective optimization is to aggregate objectives via a weighted sum:

$$m^* = \arg\max_{m \in \mathcal{M}} \sum_i w_i \, f_i(m), \tag{3}$$

where $w_i$ denotes the weight of the $i$-th objective. This formulation captures the multi-objective nature of materials discovery, where competing property goals must be jointly optimized under domain-specific constraints. The task, therefore, consists of efficiently identifying candidate materials that meet all property requirements while maximizing overall performance across objectives.

### 2.2 HYPOTHESIS GENERATION

As illustrated in Figure 1, LLEMA begins with the generation of material candidates, followed by the prediction of physicochemical properties, the evaluation of candidates and the evolutionary refinement. This stage combines the generative capabilities of LLMs with domain-guided constraints to synthesize chemically plausible hypotheses aligned with quantitative property targets. At each iteration $n$, the LLM $\pi_\theta$ samples a batch of $b$ candidate materials $\mathcal{M}^b$ from the prompt $\mathbf{p}_n$. Appendix D.2, Figure 6 details the construction of this prompt, composed of four components:

- **Task Specification:** The task description contains the natural language discovery objective $\mathcal{T}$ (e.g., "wide-bandgap semiconductors") ensuring that candidate generation remains aligned with the overarching goals of the task, while also encoding the property constraints $\mathcal{C}$ (e.g., band gap $\geq 2.5$ eV, formation energy $\leq -1.0$ eV/atom) that distinguish valid from invalid designs.
- **Chemistry-Informed Design Principles:** After the initial generation ($n = 0$), the prompt incorporates chemistry-informed design rules $\mathcal{R}$ (e.g., same-group elemental substitutions, stoichiometry-preserving replacements). These rules act as operators that encode domain knowledge, guiding the search toward chemically meaningful regions of the space while maintaining enough flexibility to allow for novel discoveries. Appendix B.4 details the evolution rules.
- **Demonstrations:** The population buffer $\mathcal{P}_{n-1}$ maintains separate buffers to hold examples of successful ($\mathbb{M}^+$) and failed ($\mathbb{M}^-$) candidates from prior iterations. Storing them in distinct buffers makes it possible to supply balanced demonstrations in $\mathbf{p}_n$, providing the LLM with explicit in-context feedback. This organization helps the model infer decision boundaries between promising and invalid designs more effectively.
- **Crystallographic Representation:** For each proposed candidate $\mathcal{M}_j$, the LLM outputs a crystallographic configuration in structured JSON format, specifying the reduced chemical formula, lattice parameters, atomic species, and fractional coordinates (Figure 1B). This standardized, machine-readable representation enables direct downstream evaluation with the property-predictor $f$, which predicts physicochemical properties and updates the population state.

## 2.3 PHYSICOCHEMICAL PROPERTY PREDICTION

Following candidate generation, LLEMA estimates the physicochemical properties of the CIF representation of each material using a hierarchical prediction system. For a given candidate $\mathcal{M}_j$, the workflow first queries the reference model, which retrieves property values from curated experimental and computational databases like Materials Project (Jain et al., 2013) through exact or similarity-based matching. For out-of-distribution candidates, which lie outside the coverage of this reference model, LLEMA employs surrogate models such as CGCNN (Xie & Grossman, 2018) and ALIGNN (Choudhary & DeCost, 2021) to provide predictions. This yields a property vector $f(m) \in \mathbb{R}^d$, where each component corresponds to a physicochemical attribute of interest. The vector is subsequently evaluated by a multi-objective scoring function against the design constraints $\mathcal{C}$. Additional details on the implementation of these surrogate models are provided in Appendix B.1.

## 2.4 FITNESS ASSESSMENT AND MEMORY MANAGEMENT

Candidates are evaluated using a multi-objective scoring function that measures the degree of alignment between their predicted properties and the target design constraints $\mathcal{C}$. For each candidate material $\mathcal{M}_j$ generated in iteration $n$, the set of predicted properties is denoted by $f_i(\mathcal{M}_j)_{i=1}^k$. The composite score is then computed as

$$S(\mathcal{T}, \mathcal{C}; \mathcal{M}_j) = \sum_{i=1}^{k} w_i \cdot \Phi_i(f_i(\mathcal{M}_j), c_i); \tag{4}$$

where $w_i$ represents the relative importance of the $i$-th property, $c_i$ denotes the corresponding target constraint, and $\Phi_i(\cdot, \cdot)$ is a normalized reward function that quantifies the satisfaction of the constraint $c_i$ by the predicted value $f_i(\mathcal{M}_j)$ (See Appendix B.2). Each candidate is then assigned to one of two memory pools: the success pool $\mathbb{M}^+$, containing candidates that satisfy all hard constraints (i.e., $S \geq 0$; $\Phi_i \geq 0$ for all $i$), and the failure pool $\mathbb{M}^-$, containing candidates that violate one or more constraints. To progressively improve candidate quality and guide the search toward property-compliant regions, LLEMA employs a memory-based evolutionary refinement step inspired by island-model strategies (Romera-Paredes et al., 2024; Shojaee et al., 2025a; Abhyankar et al., 2025). The candidate population is divided into $m$ independent islands containing success ($\mathbb{M}^+$) and failure memory ($\mathbb{M}^-$), each initialized with a copy of the initial exemplars. This structure supports parallel exploration, enabling different regions of the chemical space to evolve independently and explore a range of candidates. At each iteration $n$, we first select one of the $m$ ($m = 5$) islands using Boltzmann sampling (De La Maza & Tidor, 1992), with a score-based probability of choosing a cluster $i$: $P_i = \frac{exp(s_i/\tau_c)}{\sum_j exp(s_j/\tau_c)}$, where $s_i$ denotes the mean score of the $i$-th cluster and $\tau_c$ is the

temperature parameter. Within the chosen island, candidates are sampled from memory to construct the next prompt $\mathbf{p}_{n+1}$. Specifically, top-$k$ selection is applied to $\mathbb{M}^+$ and $\mathbb{M}^-$ to provide explicit demonstrations of high-scoring exemplars along with constraint violations. This mixture of successful and unsuccessful candidates, combined with domain-specific evolution rules $\mathcal{R}$, forms the in-context examples that guide the LLM in generating new candidates.

As described in Algorithm 1, we initialize the population of materials $\mathcal{P}_0$ and the candidate pool $\mathcal{M}$ that stores feasible solutions under the given design constraints $\mathcal{C}$. Subsequently, we construct a prompt by combining the task description with the associated physicochemical design constraints for generating candidate materials. At each iteration $n$, top-$k$ entries from prior iterations (from one of the $m$ islands), along with pre-defined evolution rules $\mathcal{R}$ (e.g., stoichiometry, oxidation state, substitution rules) provided by domain scientists, are injected to ensure chemical and structural validity while exploring the candidate space. The LLM uses the prompt to generate crystallographic information file (CIF)–based material representations. These candidates are then evaluated using an oracle predictor to estimate their physicochemical properties. A scoring function assesses their performance relative to the target design objectives, partitioning candidates into success or failure pools based on whether they satisfy the constraints. Memory is updated accordingly and a balanced sampling from both success and failure trajectories is used to provide feedback to the LLM, ensuring both exploitation of high-performing regions and exploration of underexplored design spaces. This iterative loop continues for $N$ rounds, after which the optimized candidate set $\mathbb{M}^+$ is returned.

---

**Algorithm 1** LLEMA

**Require:** Task $\mathcal{T}$, Design constraints $\mathcal{C}$, Evolution rules $\mathcal{R}$, Predictor $f$, LLM $\pi_\theta$, iterations $N$
**Ensure:** Optimized material population $\mathbb{M}^+$
    ▷ Initialize population
1: $\mathcal{P}_0 \leftarrow \texttt{InitPop()}$
    ▷ Initialize candidate pool
2: $\mathcal{M} \leftarrow \texttt{InitCand()}$
3: $\mathbf{p} \leftarrow \texttt{BuildPrompt}(\mathcal{T}, \mathcal{C})$
4: **for** $n = 1$ **to** $N - 1$ **do**
    ▷ Add rules and population data
5:    $\mathbf{p}_n \leftarrow \mathbf{p} + \mathcal{P}_{n-1}.\texttt{topk()} + \mathcal{R}$
    ▷ Crystallographic molecule generation
6:    $\mathcal{M}_{j=1}^b \leftarrow \pi_\theta(\mathbf{p}_n)$
7:    **for** $j = 1$ **to** $b$ **do**
    ▷ Physicochemical property prediction
8:       $\lambda_j = f(\mathcal{M}_j)$
    ▷ Evaluation and population update
9:       **if** $\lambda_j \in \mathcal{C}$ **then**
10:         $\mathbb{M}^+ \leftarrow \mathcal{M}_j, \lambda_j$
11:       **else**
12:         $\mathbb{M}^- \leftarrow \mathcal{M}_j, \lambda_j$
13:       **end if**
14:       $\mathcal{P}_n \leftarrow \texttt{UpdatePop}(\mathcal{P}_{n-1}, \mathbb{M}^+, \mathbb{M}^-)$
15:    **end for**
16: **end for**
17: **Return:** $\mathbb{M}^+$

---

### 2.5 IMPLEMENTATION DETAILS

To evaluate a new task, the user is only required to supply a CSV file specifying the task name and associated property constraints. From this input, the agentic framework automatically and iteratively constructs prompts to generate candidate materials, followed by their crystallographic structures in CIF format using an LLM. Next, for property evaluation, LLEMA adopts a hierarchical oracle strategy where the candidates are first queried against curated materials databases, while pretrained ML surrogates (e.g., CGCNN, ALIGNN) are invoked only for out-of-distribution compounds. These surrogates are used in inference mode with publicly available pretrained weights, which avoids retraining and significantly reduces computational overhead. For reproducibility, we report all surrogate model checkpoints and APIs in Table 5 (Appendix B.1). Candidates violating hard constraints are directly assigned low scores, ensuring efficient pruning before expensive evaluations. A detailed implementation of LLEMA is provided in Appendix D.2.

## 3 EXPERIMENTS

### 3.1 DATASETS AND BENCHMARKS

We evaluate LLEMA on fourteen application-driven discovery tasks spanning electronics, energy, coatings, optics, and aerospace, to probe multi-objective reasoning under realistic constraints, and thermodynamic stability (Table 2). Each task reflects an industrially relevant challenge and is designed to probe multi-objective reasoning under realistic constraints. Our benchmark design follows three guiding principles: (i) **Application relevance:** the target properties align with pressing technological needs such as sustainable energy, advanced electronics, and structural resilience; (ii) **Multi-constraint optimization:** tasks involve simultaneous optimization of multiple, often competing, objectives (e.g.,

maximizing hardness and conductivity), mirroring real-world engineering specifications; and (iii) **Thermodynamic stability:** all tasks enforce stability requirements through energy-above-hull criteria ($E_{hull} \approx 0$) to ensure synthetic accessibility rather than purely theoretical feasibility. Appendix C contains additional details on all tasks, including property thresholds and predictive models.

**Table 2: Materials Discovery Benchmark.** Each task is characterized by its application domain and quantitative property constraints.

| Task | Domain | Property Constraints |
|---|---|---|
| Wide-Bandgap Semiconductors | Electronics | Band gap $\geq 2.5\,\text{eV}$
Formation energy $\leq -1.0\,\text{eV/atom}$ |
| SAW/BAW Acoustic Substrates | Acoustics / Optics | Shear modulus 25–150 GPa
Dielectric constant 3.7–95 |
| High-$k$ Dielectrics | Dielectrics | Dielectric constant 10–90; Band gap 2.5–6.5 eV |
| Solid-State Electrolytes | Energy | Formation energy $\leq -1.0\,\text{eV/atom}$; Band gap $\geq 2.0\,\text{eV}$
Must contain Li, Na, K, Mg, Ca, or Al |
| Piezo Energy Harvesters | Energy | Piezoelectric coefficient $\geq 8\,\text{pC/N}$; $10 \leq \kappa \leq 8000$
(If ranking: use FoM $d^2/\kappa$; higher is better) |
| Transparent Conductors | Electronics | Band gap $> 3.0\,\text{eV}$; $50 \leq$ Electrical conductivity $\leq 5000\,\text{S/cm}$
Must be thermodynamically stable; |
| Insulating Dielectrics | Dielectrics | Band gap $\geq 2.5\,\text{eV}$; Dielectric constant $\geq 8.0$ |
| Photovoltaic Absorbers | Energy | Band gap 0.7–2.0 eV; Formation energy $\leq 0.0\,\text{eV/atom}$
Earth-abundant and non-toxic elements only |
| Hard Coating Materials | Mechanical | Bulk modulus $\geq 200\,\text{GPa}$; Formation energy $\leq -1.0\,\text{eV/atom}$; Band gap $\geq 3.0\,\text{eV}$ |
| Hard, Stiff Ceramics | Structural | Bulk modulus 100–300 GPa; Shear modulus 60–200 GPa |
| Structural Materials for Aerospace | Aerospace | Density $\leq 5.0\,\text{g/cm}^3$; Bulk modulus $\geq 100\,\text{GPa}$
Shear modulus $\geq 40\,\text{GPa}$; Energy above hull $\leq 5.0\,\text{eV/atom}$ |
| Acousto-Optic Hybrids | Acoustics / Optics | $3 \leq$ Piezoelectric coefficient $\leq 9\,\text{pC/N}$; $8 \leq \kappa \leq 85$ |
| Low-Density Structures | Aerospace | Density $\leq 3.5\,\text{g/cm}^3$
$65 \leq$ Shear modulus $\leq 195\,\text{GPa}$ |
| Toxic-Free Perovskite Oxides | Electronics / Sustainability | Band gap $\geq 2.0\,\text{eV}$; $90 \leq$ Bulk modulus $\leq 135\,\text{GPa}$
Exclude Pb, Cd, Hg, Tl, Be, As, Sb, Se, U, Th; Prefer stable $ABO_3$ oxides |

## 3.2 EXPERIMENTAL SETUP

We assessed LLEMA using complementary evaluation criteria designed to capture both the efficiency and the quality of material generation. **Hit-Rate** measures the percentage of generated candidates that simultaneously satisfy all property constraints, quantifying the efficiency of valid discovery. **Stability** evaluates the percentage of valid and thermodynamically stable, reflecting physical practicality beyond theoretical feasibility. Specifically, materials having an energy above the hull value less than $0.1$ eV/atom are considered stable. Finally, **Pareto Front Analysis** compares the quality of multi-objective trade-offs, with superior methods producing non-dominated solutions that span larger and more diverse regions of the design space. We benchmark LLEMA against generative models such as **CDVAE** (Xie et al., 2022), **G-SchNet** (Gebauer et al., 2019), **DiffCSP** (Jiao et al., 2023), and **MatterGen** (Zeni et al., 2025), as well as LLM-driven approaches including **LLMatDesign** (Jia et al., 2024) and direct prompting baselines. For comparison, all non-LLM baselines (e.g., CDVAE, G-SchNet, DiffCSP) were assigned $\sim 10\times$ candidate generations than LLM-based methods, generating 1500 samples, to offset their lack of in-context feedback. Unlike the baselines, LLEMA refines its outputs by iteratively sampling candidates from its experience buffer through an in-context refinement process. Detailed implementation settings for baselines are provided in Appendix D.

## 3.3 QUANTITATIVE RESULTS

Table 3 reports task-specific performance across fourteen benchmark domains, measured by hit-rate (H.R) and stability (Stab.) under varying physical and chemical constraints. Overall, LLEMA consistently outperforms all baselines, achieving higher hit-rates and markedly better stability across diverse material classes. Traditional generative models perform reasonably well on tasks like 'hard, stiff ceramics' or 'acousto-optic hybrids' but often produce valid yet unstable candidates. Even LLM-based baselines show limited robustness, performing well in some domains but failing in others, suggesting poor generalization across different physical regimes. In contrast, LLEMA consistently achieves higher hit-rates and markedly greater stability than the baselines. This elevated

stability indicates that evolutionary refinement and chemistry-informed rules enable LLEMA not only to meet design constraints more reliably but also to generate thermodynamically consistent and physically meaningful structures. Overall, these results demonstrate that LLEMA generalizes robustly across material classes, combining exploration and constraint satisfaction more effectively than prior methods. A deeper discussion on the qualitative aspects of LLEMA follows in the upcoming sections.

**Table 3: Comparison of Baselines on Materials Discovery Benchmark.** We implemented LLEMA with `GPT-4o-mini` and `Mistral-Small-3.2-24B-Instruct-2506`, against state-of-the-art baselines across materials design tasks. We report hit-rate (H.R.) and stability (Stab.), where higher values indicate better performance.

| Method | Wide-Bandgap Semicond. | | SAW/BAW Acoustic Substrates | | High-$k$ Dielectrics | | Solid-State Electrolytes | | Piezo Energy Harvesters | | Transparent Conductors | | Insulating Dielectrics | |
|---|---|---|---|---|---|---|---|---|---|---|---|---|---|---|
| | H.R | Stab. | H.R | Stab. | H.R | Stab. | H.R | Stab. | H.R | Stab. | H.R | Stab. | H.R | Stab. |
| CDVAE | 0.04 | 0.04 | 0.29 | 0.00 | 0.82 | 0.00 | 0.04 | 0.04 | 42.19 | 0.00 | 0.00 | 0.00 | 1.06 | 0.12 |
| G-SchNet | 0.00 | 0.00 | 0.42 | 0.00 | 0.00 | 0.00 | 0.00 | 0.00 | 0.01 | 0.00 | 2.49 | 0.00 | 0.01 | 0.00 |
| DiffCSP | 0.00 | 0.00 | 0.36 | 0.00 | 0.75 | 0.00 | 0.00 | 0.00 | 41.21 | 0.00 | 0.01 | 0.00 | 1.13 | 0.04 |
| MatterGen | 6.56 | 4.15 | 26.27 | 0.00 | 0.64 | 0.00 | 5.33 | 3.11 | 21.64 | 0.00 | 9.38 | 0.00 | 0.91 | 0.10 |
| End2end | 0.95 | 0.79 | 10.32 | 0.65 | 0.00 | 0.00 | 0.49 | 0.30 | 10.34 | 0.28 | 0.00 | 0.00 | 0.00 | 0.00 |
| LLMatDesign | 4.19 | 1.13 | 47.59 | 0.13 | 1.35 | 0.32 | 2.51 | 2.44 | 32.16 | 1.38 | 0.04 | 0.04 | 0.21 | 0.08 |
| **LLEMA (Mistral)** | 17.08 | 10.71 | 31.58 | 6.80 | 7.53 | 3.62 | 31.79 | 20.78 | **67.11** | **4.84** | **43.87** | **18.48** | **21.54** | **9.42** |
| **LLEMA (GPT)** | **33.62** | **22.42** | **59.88** | **10.74** | **19.96** | **12.68** | **46.17** | **25.37** | 63.46 | 3.22 | 39.11 | 14.85 | 17.64 | 4.60 |

| Method | Photovoltaics Absorbers | | Hard Coating Materials | | Hard, Stiff Ceramics | | Aerospace Materials | | Acousto-optic Hybrids | | Low Density Structures | | Perovskite Oxides | |
|---|---|---|---|---|---|---|---|---|---|---|---|---|---|---|
| | H.R | Stab. | H.R | Stab. | H.R | Stab. | H.R | Stab. | H.R | Stab. | H.R | Stab. | H.R | Stab. |
| CDVAE | 1.07 | 0.00 | 0.00 | 0.00 | 15.25 | 0.11 | 1.18 | 0.00 | 21.85 | 0.00 | 0.00 | 0.00 | 0.00 | 0.00 |
| G-SchNet | 0.00 | 0.00 | 0.00 | 0.00 | 0.20 | 0.20 | 0.06 | 0.00 | 0.01 | 0.00 | 0.17 | 0.00 | 0.04 | 0.00 |
| DiffCSP | 1.11 | 0.00 | 0.00 | 0.00 | 14.75 | 0.00 | 0.09 | 0.00 | 21.53 | 0.01 | 0.00 | 0.00 | 0.04 | 0.00 |
| MatterGen | 1.88 | 0.00 | 2.12 | 0.00 | 8.23 | 0.00 | 7.34 | 0.00 | 11.24 | 0.00 | 0.27 | 0.00 | 0.93 | 0.00 |
| End2end | **24.59** | **10.72** | 0.00 | 0.00 | 14.27 | 5.13 | 0.00 | 0.00 | 8.57 | 0.64 | **1.99** | **0.40** | 0.00 | 0.00 |
| LLMatDesign | 3.92 | 0.00 | 0.00 | 0.00 | 19.00 | 0.41 | 0.00 | 0.00 | 15.45 | 0.55 | 0.07 | 0.00 | 1.10 | 0.81 |
| **LLEMA (Mistral)** | 20.47 | 3.71 | 10.80 | 1.42 | 27.92 | 2.65 | **1.50** | **0.54** | 14.04 | 0.50 | 1.51 | 0.14 | **22.90** | **2.78** |
| **LLEMA (GPT)** | 22.90 | 4.76 | **17.78** | **4.61** | **30.99** | **5.73** | 0.97 | 0.26 | **26.26** | **0.82** | 0.47 | 0.14 | 19.37 | 2.79 |

### 3.4 QUALITATIVE RESULTS

We analyzed qualitative outcomes to understand LLEMA's behavior under realistic discovery settings, focusing on: (i) **Convergence dynamics**, which examines how iterative feedback progressively steers the LLM toward feasible design regions; (ii) **Pareto trade-offs**, which assess whether the method can balance competing objectives under strict property constraints; and (iii) **Discovered candidates**, which illustrate how novel yet chemically plausible compositions emerge and how they align with domain knowledge.

**Convergence Toward Feasible Frontiers.** First, we analyzed how solutions evolve over iterations. Early generations scatter broadly across the search space, often violating property constraints or collapsing into suboptimal regions. As the search progresses, the feedback from the oracle, evolutionary memory, and constraint-based selection gradually steers LLEMA toward more promising areas of chemical design space. The proportion of valid candidates increases from roughly 27% at the 250th iteration to about 33% near the 1000th iteration, indicating that the algorithm effectively exploits high-fitness regions while maintaining exploration across diverse chemical configurations. This gradual improvement reflects stronger selection pressure toward feasible trade-offs, resulting in a visible migration of candidate clusters into constraint-satisfying zones (e.g., band gaps above 2.5 eV with low formation energies). These convergence dynamics show how iterative feedback-driven evolution

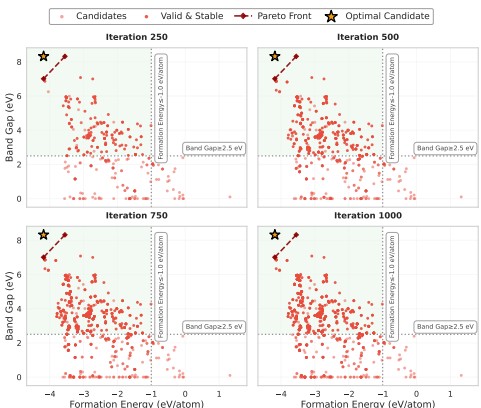

**Figure 2:** Evolution of candidates for the Wide-Bandgap Semiconductor task at different stages of LLEMA.

refines the LLM's proposal distribution, preserving diversity while increasingly focusing on candidates that balance multiple objectives in physically meaningful ways.

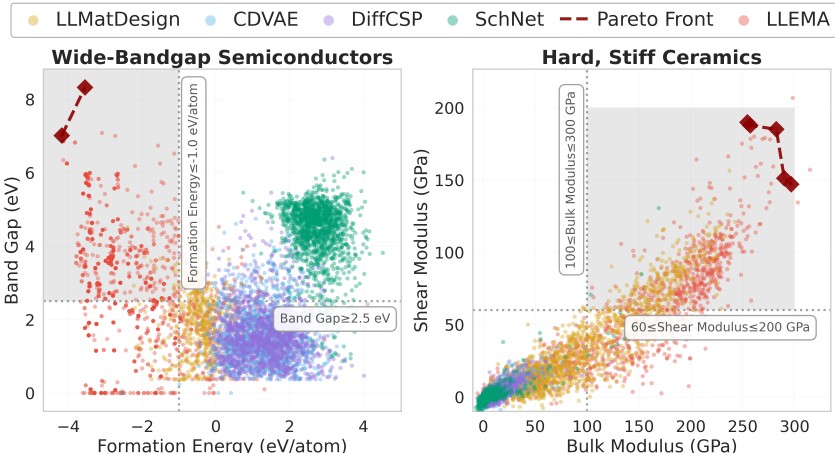

**Figure 3:** Pareto front analysis of candidate materials for two design tasks. (a) Wide-Bandgap Semiconductors; (b) Hard, Stiff Ceramics.

**Pareto Tradeoff.** We then examined the Pareto fronts for the *Wide-Bandgap Semiconductor* and *Hard–Stiff Ceramic* tasks, both of which impose stringent property requirements: semiconductors must exhibit band gaps $\geq 2.5$ eV with formation energies $\leq -1.0$ eV/atom to ensure functionality and stability, while ceramics require bulk moduli in the range $100$–$300$ GPa and shear moduli between $60$–$200$ GPa. As shown in Figure 3, the optimal Pareto front is completely dominated by LLEMA, and all Pareto-optimal solutions originate from our method on both tasks. This shows that LLEMA not only generates a higher proportion of valid and thermodynamically stable candidates, but also consistently identifies the globally optimal trade-offs between competing objectives. By enforcing explicit constraints and applying domain-guided evolutionary refinement, LLEMA effectively filters infeasible solutions while converging toward the true Pareto-optimal frontier, surpassing all baseline methods that fail to reach this region of the design space.

**Discovered Candidates.** Finally, we evaluated the plausibility of discovered compositions in the real world. We find that the materials proposed by LLEMA align with families previously investigated by domain experts, underscoring their ability to generate realistic candidates. For example, in the High-$k$ dielectric task, LLEMA suggests $ZrAl_2O_5$ and $Hf_{0.5}Zr_{0.5}O_2$, which connect closely to Zr–Al and Hf–Zr oxides studied as promising high-$k$ materials (Hakala et al., 2006; Das & Jeon, 2020; Islam et al., 2021). Similarly, LLEMA reflects expert strategies, such as substitution and doping, e.g., proposing BaHfZr oxide, which is consistent with known dopant-driven improvements in HfZr oxides (Kim et al., 2024). In photovoltaics, candidates such as CaZnSi and MgZnSi oxides emerge, which, although not directly reported, are chemically related to established ZnO-based systems (Esgin et al., 2022). These examples demonstrate that LLEMA not only respects constraints, but also uncovers novel yet chemically plausible families, validating its utility to guide real-world discovery.

## 4 ANALYSIS

### 4.1 IMPACT OF DOMAIN-GUIDED EVOLUTIONARY REFINEMENT

We evaluate the benefits of evolutionary refinement with domain-guided generation rules by benchmarking LLEMA against two baselines: (i) an LLM with iterative feedback (LLM *w/* Memory), and (ii) an unguided mutation–crossover search implemented within a multi-island evolutionary framework following Romera-Paredes et al. (2024). All methods were run for 250 iterations across four benchmark datasets, with property constraints

**Table 4:** Comparison of hit-rate (H.R), stability (Stab.), and Memorization Rate (Mem.) across generation methods aggregated over four tasks.

| Method | H.R↑ | Stab.↑ | Mem.↓ |
|---|---|---|---|
| LLM | 4.4 | 1.8 | 95.3 |
| *w/* Memory | 15.1 | 20.1 | 58.3 |
| *w/* Mutation & Crossover | 29.8 | 21.5 | 25.3 |
| **LLEMA** | **30.2** | **27.6** | **16.6** |

relaxed by 20% to allow broader exploration while preserving task relevance. The results of Table 4

show that LLMs with iterative feedback yield limited improvement, as the model tends to recall known materials and overfit to training patterns. However, the introduction of multi-island evolution substantially improves hit-rate and stability by promoting parallel exploration and mitigating premature convergence, though the absence of chemical constraints results in memorization. Incorporating chemistry-informed generation rules in LLEMA achieves the best overall balance (H.R = 30.2, Stab. = 27.6, Mem. = 16.6), constraining the search to thermodynamically and compositionally plausible regions while maintaining diversity. This staged refinement from the use of memory to the evolutionary search and domain-guided evolution demonstrates how each component progressively enhances exploration, stability, and chemical realism in generative materials design.

## 4.2 MEMORIZATION VS. GUIDED EXPLORATION

A central challenge in leveraging large language models (LLMs) for scientific discovery is their tendency to memorize training data and regenerate it during generation, rather than exploring novel solutions. Prior work has shown that LLMs frequently reproduce examples from their training corpus (Carlini et al., 2021; Hartmann et al., 2023). In materials discovery, this manifests as repeated suggestions of compounds already present in databases such as the Materials Project, leading to high duplication rates and limited novelty. We compare three approaches: a direct LLM call, an LLM augmented with a memory buffer and iterative feedback, and LLEMA. The direct LLM call exhibits the highest duplication and near-total reliance on the Materials Project (e.g., High-$k$ dielectrics show almost 100% overlap). Although adding a memory buffer helps the model explore diverse leads, it still shows a high rate of memorization, thus implying that simply storing past candidates does not guide the search toward new or diverse regions of chemical space. In contrast, incorporating a multi-island evolutionary framework enables the model to avoid local optima and repeated patterns. When further combined with chemically informed rules such as oxidation-state consistency, stoichiometry preservation, and prototype substitution, LLEMA effectively reduces redundancy and expands exploration into novel, chemically plausible regions of the design space. Together, these components push the model beyond memorization toward genuine discovery.

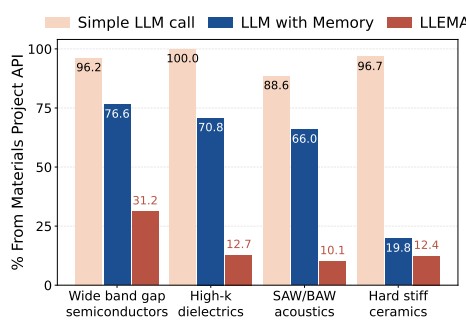

**Figure 4:** Percentage of generated candidates from the Materials Project across four domains for different baselines. Lower values indicate less memorization.

## 4.3 IMPACT OF SURROGATE MODELS

Figure 5 quantifies the effect of surrogate model–based property prediction on LLEMA's performance. All experiments were run for 250 iterations with task constraints relaxed to encourage exploration, similar to Section 4.1. To isolate the contribution of ML-based surrogates, we removed surrogate ML models like CGCNN and ALIGNN, and restricted the workflow to using only the Materials Project database for property annotations. Furthermore, we experimented with fewer iterations while relaxing task constraints for the wide-bandgap semiconductors dataset. Even in this setting, both hit-rate and stability collapse to near-zero ($< 5\%$), as the evolutionary process cannot assign meaningful rewards without surrogate models to candidates for missing property annotations in the Materials Project API. The search then drifts toward trivial or repeated compounds instead of progressing toward novel solutions. By contrast, reintroducing surrogate predictors produces more than a sixfold improvement, increasing hit-rate and stability into the 25–30% range. The surrogate estimates for the out-of-distribution candidates provide the evaluation signals needed to sustain exploration beyond the sparse coverage of existing datasets.

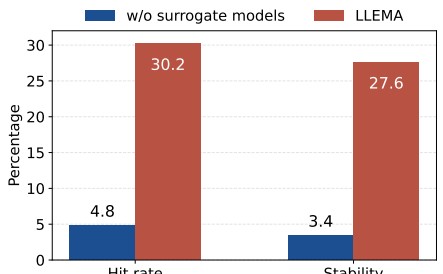

**Figure 5:** Hit-Rate and Stability performance of LLEMA with and without surrogate model-based property prediction.

These results demonstrate that surrogate models are indispensable for providing informative fitness signals under sparse supervision, preventing collapse, and enabling effective discovery.

## 5  RELATED WORK

**Material Discovery.**   Materials discovery has progressed from trial-and-error (Nagamatsu et al., 2001) to *ab initio* modeling (Jain et al., 2011; Hautier et al., 2012; Pyzer-Knapp et al., 2015), with DFT and high-throughput screening as standard tools. Machine learning further accelerates discovery via rapid property prediction (Xie & Grossman, 2018; Chen et al., 2019; Choudhary & DeCost, 2021) and generative design (Gebauer et al., 2019; Xie et al., 2022; Jiao et al., 2023), including multi-objective optimization (Gopakumar et al., 2018; Jablonka et al., 2021). Yet, these methods remain limited by data scarcity and poor transferability across domains. In contrast, LLMs have broad scientific knowledge that enables reasoning in data-poor regimes, making them a natural foundation for knowledge-guided materials design frameworks.

**LLMs and Evolutionary Algorithms.**   Recent advances in generative models have shown their ability to generalize across diverse tasks using pre-trained knowledge and simple prompting strategies (Brown et al., 2020; Reddy & Shojaee, 2025; Wei et al., 2022). However, their outputs are often unreliable or inconsistent (Madaan et al., 2024; Zhu et al., 2023), motivating the use of evolutionary optimization frameworks where LLMs act as generators and external evaluators guide selection and refinement (Lange et al., 2024; Lehman et al., 2023; Liu et al., 2024; Zheng et al., 2023). Such frameworks have been successfully applied in areas including *code and prompt generation* (Guo et al., 2024), *scientific discovery* (Shojaee et al., 2025a;b), *mathematical optimization* (Yang et al., 2024b), *program synthesis* (Romera-Paredes et al., 2024), *robotics reward design* (Ma et al., 2024), *feature engineering* (Abhyankar et al., 2025), and *chemical discovery* (Wang et al., 2025). We extend this line of work to real-world materials science challenges by incorporating chemistry-informed LLM evolution that enforces structural validity and physical plausibility, enabling principled discovery under multi-objective constraints.

**LLMs in Material Science.**   Early work applied LLM-based frameworks to literature mining, named-entity recognition, and property extraction (Gupta et al., 2022; 2024; Niyongabo Rubungo et al., 2025). Beyond text extraction, more recent methods use LLMs for hypothesis generation (Miret & Krishnan, 2024), synthesis route planning, and as reasoning engines in multi-agent systems (Zhang et al., 2024; Kang & Kim, 2024; Kumbhar et al., 2025). Systems such as MatAgent (Bazgir et al., 2025) and LLMatDesign (Jia et al., 2024) combine LLM reasoning with property predictors and optimization loops, but often emphasize narrow objectives and weak constraint enforcement, leading to an unguided or infeasible search. Closely related efforts, such as MatLLMSearch (Gan et al., 2025), use LLM proposals in an evolutionary refinement loop for crystal discovery, and MatterGen (Zeni et al., 2025) uses a diffusion-based model with conditional sampling for the inverse design. Although effective, these approaches rely on either learned generative priors or limited objective formulations. LLEMA instead frames the design of materials as a multi-objective search problem with synthesizability, integrating LLM reasoning with surrogate predictors, domain constraints, and memory-based refinement to balance novelty, feasibility and property alignment.

## 6  CONCLUSION

In this work, we present LLEMA, a unified framework that integrates the evolutionary paradigm, domain-guided rules, and the scientific knowledge of LLMs to enable multi-objective and synthesizability-aware materials discovery. To rigorously evaluate generality, we curate a benchmark suite of 14 diverse, real-world discovery tasks spanning electronics, energy, coatings, optics, and aerospace. Our experiments demonstrate three key findings: (i) LLEMA achieves *higher hit rates and stronger Pareto fronts* than generative and LLM-only baselines, showing improved ability to balance competing design objectives. (ii) LLEMA produces a larger fraction of *thermodynamically stable and chemically plausible compounds*, validating its emphasis on synthesizability. (iii) LLEMA significantly *reduces duplication and corpus recall*, mitigating the memorization tendency of vanilla LLM generation and enabling genuine exploration of novel chemical space. While these results highlight the potential of LLEMA, our reliance on surrogate predictors, limited experimental validation, and the cost of iterative LLM queries suggest opportunities for future work paving the way toward scalable and reliable automated materials discovery.

ACKNOWLEDGEMENTS

This research was partially supported by the U.S. National Science Foundation (NSF) under Grant No. 2416728 and Autodesk Research. The authors thank Modal for providing computational resources that supported the hosting and implementation of the models used in this study. Saaketh Desai is supported in part by the Center for Integrated Nanotechnologies, an Office of Science user facility operated for the U.S. Department of Energy. This article has been authored by an employee of National Technology & Engineering Solutions of Sandia, LLC under Contract No. DE-NA0003525 with the U.S. Department of Energy (DOE). The employee owns all rights, title, and interest in and to the article and is solely responsible for its contents. The United States Government retains, and the publisher, by accepting the article for publication, acknowledges that the United States Government retains a non-exclusive, paid-up, irrevocable, worldwide license to publish or reproduce the published form of this article or allow others to do so, for United States Government purposes. The DOE will provide public access to these results of federally sponsored research in accordance with the DOE Public Access Plan.

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

## A    AUTONOMOUS MATERIALS DISCOVERY

The discovery of novel materials with tailored properties is fundamental to technological progress, contributing to a global materials industry generating approximately \$ 8 trillion in revenue in 2023 (McKinsey, 2024). The impact of tailored materials is particularly significant in critical domains such as energy storage Ling (2022), photovoltaics Solak & Irmak (2023), and microelectronics Refai-Ahmed et al. (2024), where advanced materials can enhance efficiency, reduce energy footprint, and improve sustainability. The grand challenge in materials discovery lies in navigating an enormous design space encompassing a vast number of material compositions and manufacturing techniques while achieving the target material structures and properties across multiple length and time scales (Oganov et al., 2019). Human intuition-based methods, along with traditional design of experiments and computational investigations, are slow and ineffective at screening this vast space of possible materials and synthesis conditions. Consequently, the field has witnessed a rapid emergence of data-driven discovery paradigms Agrawal & Choudhary (2016) and autonomous self-driving laboratories Abolhasani & Kumacheva (2023), integrating artificial intelligence, robotics, and high-throughput computation. Data-centric platforms such as the Materials Project (Jain et al., 2013) exemplify this shift, offering open high-throughput DFT databases that enable large-scale screening of candidate materials (Horton et al., 2025). Agent-based frameworks further advance this paradigm by autonomously navigating complex chemical spaces and identifying novel stable compounds (Montoya et al., 2020). Together, these systems mark a paradigm shift, transforming materials discovery from a process of human-guided trial and error to one of algorithmic intuition and autonomous exploration.

# B  ADDITIONAL DETAILS

## B.1  SURROGATE MODEL

The oracle in LLEMA is designed to provide scalable and reliable property estimation by combining external databases with pretrained surrogates. After generating crystallographic representations (CIFs) for each candidate, we first query the Materials Project API[1] to retrieve available properties. However, the API has limited coverage: many target properties, such as electrical conductivity and dielectric constants, are either missing for several materials or absent altogether, and the database itself, though large, cannot cover all generated candidates. To overcome this limitation, we conducted preliminary experiments with several pretrained models and identified **ALIGNN** (Choudhary & DeCost, 2021) and **CGCNN** (Xie & Grossman, 2018) as the most reliable surrogates across key properties. We use ALIGNN checkpoints trained on the JARVIS-DFT dataset[2] via their official implementation[3] and the official CGCNN release[4]. This design ensures a clear mapping: when properties are available in the Materials Project, we use them directly; when they are not, the most appropriate surrogate model is selected based on our prior experimentation. By integrating database queries with pretrained ML predictors, the oracle balances accuracy, scalability, and coverage, enabling consistent evaluation across all discovery tasks. Table 5 provides a mapping between the surrogate models and the source pretrained files used to predict the corresponding physicochemical properties.

**Table 5: Mapping between target properties and oracle sources.** Electrical conductivity does not have a dedicated pretrained model and is computed from the Seebeck coefficient and power factor. Density is computed directly from the structure (CIF/POSCAR).

| Property | Surrogate Model | Source |
|---|---|---|
| Band gap | CGCNN | band-gap.pth.tar |
| Formation energy | ALIGNN | jv_formation_energy_peratom |
| Bulk modulus | ALIGNN | jv_bulk_modulus_kv |
| Shear modulus | ALIGNN | jv_shear_modulus_gv |
| Dielectric constant | ALIGNN | jv_epsx |
| Piezoelectric constant | ALIGNN | jv_dfpt_piezo_max |
| Energy above hull | ALIGNN | jv_ehull |
| Density, volume | ALIGNN / CGCNN | – |
| Seebeck coefficient | ALIGNN | jv_n-Seebeck |
| Power factor | ALIGNN | jv_n-powerfact_alignn |
| Electrical conductivity | ALIGNN | – |

## B.2  FITNESS ASSESSMENT

LLEMA uses a multi-objective scoring function to evaluate candidate materials and guide the evolutionary search process. The scoring function is defined as:

$$S(\mathcal{T}, \mathcal{C}; \mathcal{M}_j) = \sum_{i=1}^{k} w_i \cdot \Phi_i(f_i(\mathcal{M}_j), c_i),$$

where $f_i(\mathcal{M}_j)$ denotes the predicted value of the $i$-th property for candidate $\mathcal{M}_j$, $c_i$ is the corresponding design constraint, and $w_i \geq 0$ is a weighting coefficient that encodes the relative importance of property $i$. The function $\Phi_i(\cdot, \cdot)$ measures the degree of satisfaction between a predicted property value and its associated constraint, and maps it to a bounded reward space. A candidate is considered *feasible if and only if* $\Phi_i \geq 0$ *for all constraints* $i$. Each constraint $c_i$ is assumed to be one of three types: (a) a lower-bound constraint $\left( f_i(\mathcal{M}_j) \geq l_i \right)$, (b) an upper-bound constraint $\left( f_i(\mathcal{M}_j) \leq u_i \right)$, or (c) an interval constraint $\left( f_i(\mathcal{M}_j) \in [l_i, u_i] \right)$. The normalized reward $\Phi_i$ is

---

[1]https://next-gen.materialsproject.org/api

[2]https://figshare.com/articles/dataset/ALIGNN_models_on_JARVIS-DFT_dataset/17005681/6

[3]https://github.com/usnistgov/alignn

[4]https://github.com/txie-93/cgcnn

defined as a signed, clipped feasibility margin that depends only on the constraint limits. Specifically, for a lower-bound constraint,

$$\Phi_i = \mathrm{clip}\left(\frac{f_i(\mathcal{M}_j) - l_i}{\max(|l_i|, 1)}, -1, 1\right),$$

for an upper-bound constraint,

$$\Phi_i = \mathrm{clip}\left(\frac{u_i - f_i(\mathcal{M}_j)}{\max(|u_i|, 1)}, -1, 1\right),$$

and for an interval constraint,

$$\Phi_i = \mathrm{clip}\left(\frac{\min\big(f_i(\mathcal{M}_j) - l_i, \; u_i - f_i(\mathcal{M}_j)\big)}{\max(|u_i - l_i|, 1)}, -1, 1\right),$$

where $\mathrm{clip}(z, -1, 1) = \max(-1, \min(1, z))$. For example, in *wide-bandgap semiconductors*, band gap is supposed to be higher than 2.5 eV whereas for formation energy, it is better to have lower values i.e. less than $-1$ eV/atom and scores are calculated accordingly. The weighting coefficients $w_i$ encode the task-specific importance of each property, allowing LLEMA to balance competing objectives during optimization. These weights act as hyperparameters, curated in consultation with domain heuristics: performance-critical properties are emphasized, while feasibility constraints (stability-related) act as secondary filters after scoring the candidates to prevent chemically implausible candidates. However, for simplicity, we assign equal weights to all the properties associated with a task. For example, in the *wide-bandgap semiconductor* task, band gap and formation energy are given equal priorities, while energy-above-hull is used to ensure stability and synthesizability. This principled scoring, along with a stability check, ensures that the discovery process reflects the physical and industrial priorities of each benchmark task, rather than being tuned arbitrarily.

## B.3 MULTI-ISLAND EXPLORATION

LLEMA utilizes the multi-island framework to evolve its candidate population. At initialization, the global population is partitioned into $k$ independent islands by evenly splitting the candidate set into $k$ subpopulations. Each island maintains its own success and failure buffers and evolves independently of the others. Given a total evolutionary budget of $T$ iterations, each island receives approximately $T/k$ iterations. Therefore, the choice of $k$ determines the balance between exploration and exploitation. A smaller $k$ allows deeper refinement per island (greater exploitation), and a larger $k$ enables more independent trajectories with shallower refinement (greater exploration). Because all islands run in parallel with distinct memory states, $k$ primarily determines how the total computational budget is distributed across parallel search paths. We evaluate three representative settings: $k = 1$ (single trajectory), $k = 5$ (moderate multi-trajectory), and $k = 10$ (highly parallel) for 250 iterations with relaxed constraints. Results for four representative tasks are shown in Table 6. Moderate island counts (for example, $k = 5$) consistently provide the best balance: they allow broad exploration while still giving each island enough iterations to refine promising regions. A very small $k$ restricts exploration, and a very large $k$ reduces the per-island refinement depth. As each island maintains its own success and failure memories and adapts independently, the trajectories diverge early and explore different portions of the design space. As long as $k$ is chosen in proportion to the available compute budget (for example, using more islands only when $T$ is sufficiently large), overall performance remains stable across tasks.

**Table 6:** Effect of island count on representative tasks.

| # Islands | Wide-Bandgap Semicond. | High-k Dielectrics | SAW/BAW Acoustics | Hard, Stiff Ceramics |
|:---:|:---:|:---:|:---:|:---:|
| 1 | 18.62 | 5.59 | 22.00 | 5.65 |
| 5 | 33.62 | 19.96 | 59.88 | 30.99 |
| 10 | 24.46 | 20.34 | 26.78 | 16.19 |

## B.4 EVOLUTIONARY GENERATION RULES

To constrain exploration and ensure chemical validity, LLEMA incorporates a set of domain-informed rules that guide the modification of candidate materials during evolutionary refinement. These rules encode chemical heuristics such as group-wise substitutions, prototype preservation, and functional analog discovery. At each iteration, they are injected into the prompt as part of the evolutionary context, ensuring that candidate modifications follow chemically plausible pathways while maintaining diversity. Below, we enumerate the rules used in this work. Each rule is presented in monospaced format to emphasize its role as a design heuristic.

1. Same-group elemental substitution: Replace each element with another from the same periodic group.

$$A_2B_3 \;\rightarrow\; C_2D_3, \quad C \in \texttt{Group(A)},\; D \in \texttt{Group(B)}$$

2. Stoichiometry-preserving substitution: Keep the formula ratios but replace with chemically similar elements.

$$A_2B_3C_4 \;\rightarrow\; D_2E_3F_4, \quad D \sim A,\; E \sim B,\; F \sim C$$

3. Oxidation state substitution: Replace elements with others having the same oxidation state.

$$A^{2+}B^- \;\rightarrow\; C^{2+}D^-$$

4. Functional group substitution: Swap one functional group with another of similar chemical behavior.

$$\texttt{R-X} \;\rightarrow\; \texttt{R-Y}, \quad X \sim Y$$

5. Crystal prototype substitution: Maintain the structural prototype (e.g., perovskite $ABX_3$) and replace elements.

$$ABX_3 \;\rightarrow\; CDY_3$$

6. Coordination geometry mutation: Change the ligand coordination number around a central atom.

$$A(L)_4 \;\rightarrow\; A(L)_6$$

7. Oxidation/reduction variant: Adjust stoichiometry for different redox configurations.

$$A_2B_3 \;\rightarrow\; A_3B_4$$

8. Surface functionalization: Add functional groups to a known material surface.

$$\texttt{ABC} \;\rightarrow\; \texttt{ABC-X}$$

9. Template-guided combinatorics: Fill in a known formula structure with compatible elements.

$$ABX_3 \;\rightarrow\; C\text{-}D\text{-}E_3$$

10. Inverse property conditioning: Generate candidates with properties conditioned on a specified target.

$$\texttt{Target:}\;\; \texttt{High Hardness} \;\Rightarrow\; A_2B$$

11. Retrosynthesis-based forward design: Suggest plausible products from precursors.

$$A + B \;\rightarrow\; C$$

12. Functional analog discovery: Replace with another compound serving the same role.

$$\texttt{A}_2\texttt{B}_3 \texttt{ (insulator) } \rightarrow \texttt{ C}_4\texttt{D}_6 \texttt{ (insulator)}$$

13. Tolerance-factor guided substitution: Replace atoms while preserving structural stability rules.

$$\texttt{ABX}_3 \rightarrow \texttt{A'BX}_3, \quad \texttt{r(A')} \approx \texttt{r(A)}$$

14. Periodicity-preserving analog search: Replace atoms while maintaining periodic trends.

$$\texttt{A}_2\texttt{B}_3 \rightarrow \texttt{C}_2\texttt{D}_3, \quad \texttt{C} \sim \texttt{A, D} \sim \texttt{B}$$

## C  Datasets and Benchmark

To evaluate multi-objective material discovery, we curate a diverse benchmark spanning 14 representative design tasks across various domains (see Table 2). Each task defines a distinct combination of physicochemical constraints that reflect practical design objectives in real-world materials engineering. Together, the benchmark captures the breadth of challenges faced in materials design, from optimizing performance–stability trade-offs to balancing mechanical, dielectric, and sustainability objectives. The resulting dataset assesses how well models can reason over complex, interdependent physical properties and generate synthesizable candidates under realistic constraints.

**Wide-Bandgap Semiconductors.**  Wide-bandgap semiconductors underpin high-power and high-frequency electronics, as well as optoelectronic applications like UV LEDs. Candidate materials must achieve a band gap $\geq 2.5\,\text{eV}$ while maintaining formation energies $\leq -1.0\,\text{eV/atom}$ and low energy-above-hull ($\leq 0.1\,\text{eV/atom}$) to ensure both performance and stability. This task challenges models to jointly balance wide electronic gaps with realistic thermodynamic feasibility, a combination critical for next-generation power electronics and photonics.

**SAW/BAW Acoustic Substrates.**  Acoustic substrates are critical for wireless communication devices, including filters and resonators. Target materials must combine shear moduli between $25$–$150\,\text{GPa}$ with dielectric constants between $3.7$–$95$, ensuring mechanical resonance with stable dielectric response. This task probes the ability to navigate trade-offs in mechanical and dielectric behavior to identify candidates for next-generation 5G/6G communication technologies.

**High-$k$ Dielectrics.**  High-permittivity dielectrics enable miniaturization in capacitors and gate oxides for semiconductor technology. Desired materials exhibit dielectric constants between $10$–$90$ and band gaps in the range $2.5$–$6.5\,\text{eV}$, ensuring both capacitance density and insulation. The task forces models to balance polarizability against leakage resistance, reflecting practical design needs in integrated circuits.

**Solid-State Electrolytes.**  Solid electrolytes promise safe, high-energy batteries by replacing flammable liquid electrolytes. Candidates must be thermodynamically stable (formation energy $\leq -1.0\,\text{eV/atom}$), electronically insulating (band gap $\geq 2.0\,\text{eV}$), and contain mobile species such as Li, Na, K, Mg, Ca, or Al. This task reflects the fundamental trade-off between chemical stability and ionic conductivity, which is central to enabling next-generation solid-state batteries.

**Piezo Energy Harvesters.**  Energy-harvesting applications demand strong electromechanical coupling and dielectric robustness. Candidates must have piezoelectric coefficients $d_{ij} \geq 8\,\text{pC/N}$ and dielectric constants in the range $10 \leq \kappa \leq 8000$. Performance is often ranked by the figure of merit $d^2/\kappa$, which rewards high piezoelectric activity while penalizing excessive dielectric loading. This task reflects the practical requirement of optimizing efficiency under electrical and mechanical constraints in self-powered devices.

**Transparent Conductors.** Transparent conducting oxides balance optical transparency with electronic conductivity for use in displays and photovoltaics. Candidates must exhibit band gaps $E_g > 3.0\,\text{eV}$ and conductivities $50 \leq \sigma \leq 5000\,\text{S/cm}$ while remaining thermodynamically stable. This dual optimization captures the key trade-off between light transmission and carrier mobility.

**Electrically Insulating Dielectrics.** Insulating dielectrics are critical for high-voltage applications requiring minimal current leakage. Materials must have band gaps $E_g \geq 2.5\,\text{eV}$ and dielectric constants $\kappa \geq 8.0$, ensuring high breakdown strength and stable polarization response. These constraints emphasize materials with strong insulation behavior and mechanical integrity under electric fields.

**Photovoltaic Absorbers.** Photovoltaic materials must absorb sunlight efficiently while remaining stable, earth-abundant, and non-toxic. Target absorbers have optimal band gaps (0.7–2.0 eV) for solar conversion, formation energies $\leq 0.0\,\text{eV/atom}$, and must exclude rare or hazardous elements. This task reflects real-world sustainability constraints, forcing models to move beyond theoretical optima and identify candidates suitable for large-scale, affordable solar deployment.

**Hard Coating Materials.** Coatings protect industrial components from wear, corrosion, and high temperatures. Desired materials exhibit high bulk modulus ($\geq 200\,\text{GPa}$), wide band gaps ($\geq 3.0\,\text{eV}$), and strong thermodynamic stability (formation energy $\leq -1.0\,\text{eV/atom}$). This task probes the ability of models to discover coatings that simultaneously resist deformation, provide electrical insulation, and remain synthesizable which is key for aerospace, tooling, and cutting-edge manufacturing.

**Hard, Stiff Ceramics.** Ceramics used in extreme environments require resistance to deformation while maintaining stiffness across broad ranges. Candidates must exhibit bulk moduli between 100–300 GPa and shear moduli between 60–200 GPa. This task emphasizes the discovery of brittle but strong materials, essential for armor, cutting tools, and high-temperature structural applications.

**Structural Materials for Aerospace.** Aerospace materials must balance light weight with mechanical resilience. This task enforces minimum stiffness (bulk modulus $\geq 100\,\text{GPa}$, shear modulus $\geq 40\,\text{GPa}$), low density ($\leq 5.0\,\text{g/cm}^3$), and sufficient thermodynamic stability (energy above hull $\leq 5.0\,\text{eV/atom}$). It challenges models to identify materials that achieve high strength-to-weight ratios while remaining manufacturable, crucial for aviation and spaceflight.

**Acousto-Optic Hybrids.** Materials for acousto-optic devices must balance piezoelectric and dielectric properties to minimize loading while enabling efficient coupling. Candidates are required to exhibit piezoelectric coefficients in the range $3 \leq d_{ij} \leq 9\,\text{pC/N}$ and dielectric constants in the range $8 \leq \kappa \leq 85$, with a preference for a narrow $\kappa$ band to reduce dielectric loading. Ranking emphasizes proximity to target $d$-bands and mid-range $\kappa$ values, highlighting the trade-off between acoustic response and dielectric stability.

**Low-Density Structural Materials.** Aerospace-grade materials must combine low density with high stiffness-to-weight ratios. Candidates are constrained to density $\rho \leq 3.5\,\text{g/cm}^3$ and shear modulus $65 \leq G \leq 195\,\text{GPa}$, with optimization targeting the ratio $G/\rho$. The task favors lightweight systems that maintain strength and creep resistance under thermal stress.

**Toxic-Free Perovskite Oxides.** Environmentally safe perovskite oxides aim to eliminate toxic elements while preserving desirable optoelectronic properties. Candidates must have band gaps $E_g \geq 2.0\,\text{eV}$ and bulk moduli $90 \leq K \leq 135\,\text{GPa}$, while excluding Pb, Cd, Hg, Tl, Be, As, Sb, Se, U, and Th. The search prioritizes thermodynamically stable $ABO_3$ structures that retain mechanical durability without compromising sustainability.

# D  IMPLEMENTATION DETAILS

## D.1  BASELINES

We compare LLEMA against several state-of-the-art materials discovery baselines, encompassing a diverse range of methodologies from traditional deep learning-based techniques to LLM-based

methods. For all baselines, we generate candidate materials using their respective official implementations, applying minimal benchmark-specific modifications when necessary. Property values for all generated candidates are computed using the same property prediction pipeline as employed in LLEMA, ensuring a consistent and fair comparison. Specifically, we implement:

**CDVAE.** `CDVAE` (Xie et al., 2022) is a conditional variational autoencoder tailored for crystal structure generation. It learns latent representations of crystals conditioned on composition, enabling the generation of valid and diverse candidate structures. We implement `CDVAE` using the official open-source repository[5] with default parameters, outputting the generated structures in CIF (Crystallographic Information File) format, along with latent embeddings stored as NumPy arrays for downstream analysis.

**G-SchNet.** `G-SchNet` (Gebauer et al., 2019) is a graph-based deep generative model for molecular and crystal structures, built on the SchNet architecture. It incrementally generates atom types and positions conditioned on the partially built structure, allowing it to capture geometric and chemical validity. We implement G-SchNet using the open-source codebase[6] with default hyperparameters for 15000 iterations. The generated samples are stored in a database, which is subsequently converted into candidate-specific CIF files.

**DiffCSP.** `DiffCSP` (Jiao et al., 2023) introduces diffusion models for crystal structure prediction (CSP). It models the distribution over atomic positions and lattice parameters via a denoising diffusion probabilistic model, enabling efficient sampling of realistic crystal structures. We implement `DiffCSP` using the official code release[7] with default settings and store the generated candidates in CIF files.

**MatterGen.** We use the official `mattergen`[8] release and checkpoints. For candidate generation, we sample batches from the released base checkpoint. Generated candidates are then evaluated using surrogate predictors and the Materials Project API to determine the candidate quality.

**End2end.** We implemented a pipeline with base LLM as `GPT-4o-mini` to generate candidates conditioned on the design task and its corresponding property constraints. The LLM operated with a sampling temperature of $\tau = 0.8$ to promote diversity while preserving structural coherence. It was explicitly instructed to output candidates in Crystallographic Information File (CIF) format, ensuring standardized structural representations suitable for subsequent validation and property evaluation using the surrogate-assisted oracle prediction.

**LLMatDesign.** `LLMatDesign` (Jia et al., 2024) leverages LLMs for material design by prompting LLMs to iteratively improve the provided material to return the material with the single target property. As `LLMatDesign` is primarily designed for single-objective optimization, we adapt its released implementation[9] for our multi-objective materials discovery benchmarks and add surrogate-augmented oracle from our pipeline for feedback. For candidate generation, we adopt their original prompt template, which provides a material's chemical formula and task-specific properties and asks the model to propose a modification that satisfies the constraints. The model selects one of four modification types among 'exchange', 'substitute', 'remove', or 'add', and outputs both the modification and a natural-language hypothesis justifying the change.

## D.2 LLEMA

**Material Design.** Figure 6 illustrates an example prompt for the **Wide-Bandgap Semiconductor** task. The prompt begins with general instructions specifying the LLM's role and objective, followed by task-specific details such as the description of the design problem, property constraints (e.g., required band gap and formation energy), and a set of previously explored candidates with their

---

[5]https://github.com/txie-93/cdvae

[6]https://github.com/atomistic-machine-learning/G-SchNet

[7]https://github.com/jiaor17/DiffCSP

[8]https://github.com/microsoft/mattergen

[9]https://github.com/Fung-Lab/LLMatDesign

associated property values sampled from the experience buffer. We then sample $b = 2$ candidate outputs at a temperature of $\tau = 0.8$, chosen to balance creativity with adherence to constraints while exploiting promising directions in the search space. To further ensure physical plausibility, we introduce rule-based constraints during the evolutionary phase (after the initial generation step, $n = 0$), which guide the LLM through the material discovery cycle and promote chemically valid candidates. We randomly sample a subset of 6 rules from the set of rules and provide it to the LLM to guide the candidate generation. By providing task descriptions, prior evaluations, and rule-based constraints in the prompt, we effectively steer the LLM toward feasible, property-aligned material candidates.

**Crystallographic Representation.** For each sampled compound, we generate a crystallographic representation in the form of a CIF (Crystallographic Information File) (see Figure 6). A CIF encodes the lattice parameters, symmetry, and atomic positions of a material, which directly determine key properties such as band gap, formation energy, and stability. Oracle models require this structure-aware representation to correctly predict thermodynamic and electronic properties, making CIF generation essential for evaluating whether candidates are both physically plausible and functionally relevant. By mapping each candidate to a valid crystallographic structure, LLEMA enables property prediction and aligns with standard materials discovery workflows.

**Data-Driven Evaluation.** We rely on oracle-assisted surrogate models to return key property values such as band gap and formation energy, which serve as the basis for evaluating each candidate. The predictions are used to assess whether generated materials satisfy the task-specific constraints. Candidates that fully meet the constraints are scored highest, while those that partially satisfy them are ranked above those that fail entirely. This data-driven evaluation ensures that generation quality is grounded in quantitative property predictions rather than heuristic filtering, making it a critical component of the discovery pipeline.

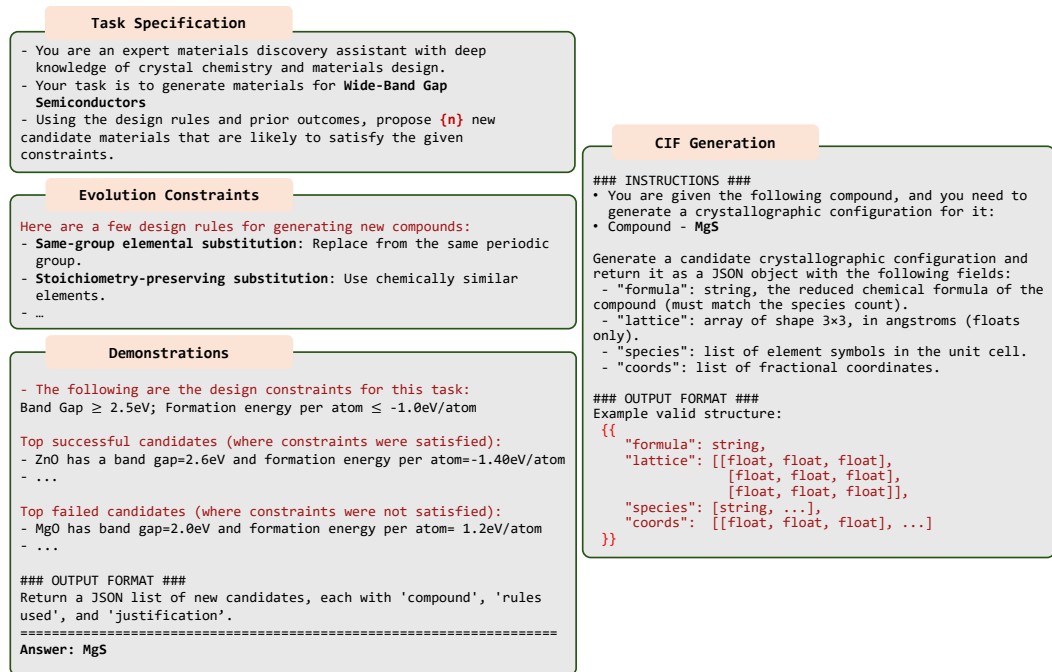

**Figure 6: Example of input prompts for the wide-bandgap semiconductors task**. (a) Candidate Generation Step , including task specification, evolution constraints, in-context demonstrations, and (b) Crystallographic File (CIF) Representation Step containing the CIF generation prompt.

**Experience Management.** We adopt an island-based evolutionary strategy (Romera-Paredes et al., 2024; Shojaee et al., 2025a; Abhyankar et al., 2025) to manage the experience buffer, where generated material candidates and their evaluation scores are distributed across $m = 5$ independently evolving

islands. Each island is initialized with a small set of seed candidates sampled from the Materials Project API and refined using the LLM. Within each island, the buffer is divided into two components: a **reward buffer**, which stores candidates that fully satisfy all task-specific constraints (e.g., band gap and formation energy), and an **error buffer**, which stores candidates that only partially satisfy or completely violate the constraints. This separation enables the framework to reinforce high-quality generations while also retaining failure cases, which serve as negative examples to guide exploration. The experience buffer is further leveraged to construct prompts for subsequent LLM calls. After the prompt template is updated with task-specific information, one of the $m$ islands is selected, and $k = 2$ candidates are sampled from its buffers to serve as in-context demonstrations. Candidate selection follows a Boltzmann strategy (De La Maza & Tidor, 1992) that assigns higher probability to clusters with stronger evaluation scores. Specifically, if $s_i$ denotes the score of the $i$-th cluster, the probability $P_i$ of selecting it is given by:

$$P_i = \frac{\exp(s_i/\tau_c)}{\sum_i \exp(s_i/\tau_c)}, \qquad \tau_c = T_0 \left(1 - \frac{u \bmod M}{M}\right),$$

where $\tau_c$ is the temperature parameter, $u$ is the current number of candidates on the island, and $T_0 = 0.1$ and $M = 10{,}000$ are hyperparameters. Once a cluster is selected, we sample candidates from it for inclusion in the next generation. This mechanism integrates information from both successful and failed generations while preserving diversity across islands, thereby guiding the LLM toward more effective material discovery.

## E  QUALITATIVE ANALYSIS

### E.1  CASE STUDY

To better understand how LLEMA's evolutionary mechanism balances exploration and exploitation, we analyze the temporal dynamics of the search process across iterations. This case study focuses on three aspects of the evolutionary trajectory: (i) **memorization rate:** the proportion of generated compounds retrieved directly from known databases such as the Materials Project; (ii) **chemical diversity:** the spread of elemental coverage across the periodic table; and (iii) **validity progression:** the fraction of syntactically and physically valid compounds discovered over successive generations.

**From Memorization to Exploration.** In the early stages of optimization, LLEMA's search behavior is dominated by memorization, with a substantial portion of generated candidates overlapping with known entries from the Materials Project (Figure 7). As the search evolves, the proportion of externally sourced structures rapidly declines. From about 83% initially, the overlap with Materials Project drops to 10% by 250 iterations, eventually dropping to about 3% by the end of the evolution. This steady decrease highlights a clear transition from memorization of existing chemical knowledge to exploration of novel compositions. By later iterations, the candidate pool is largely composed of previously unseen structures, indicating that the model has shifted from recall-driven synthesis toward genuine materials discovery. Figure 8 further highlights how the evolution process guides the exploration toward a more diverse set of elements.

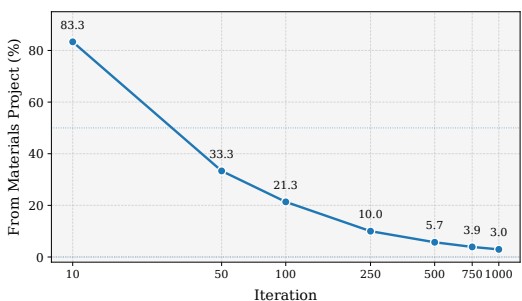

**Figure 7:** The percentage of candidate structures sourced from the Materials Project across iterations by LLEMA for SAW/BAW Acoustic Substrates.

**Convergence Toward Feasible Frontiers.** Tracking candidates in property space reveals how LLEMA's population progressively migrates toward feasible and optimal regions over time. As shown in Figure 9, at early stages (Iteration 250), only about 30% of SAW/BAW acoustic substrates satisfy physical validity constraints, with most scattering across thermodynamically unstable or

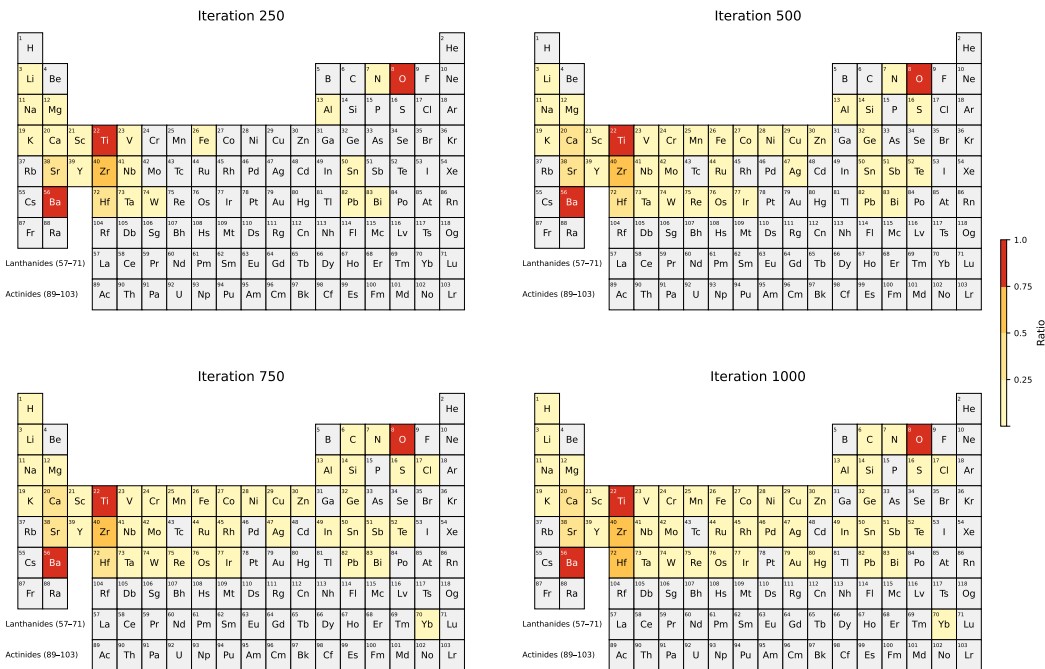

**Figure 8:** Evolution of periodic table coverage during SAW/BAW acoustic substrate optimization. Each panel shows the element-wise usage ratio across iterations (250, 500, 750, 1000) in the evolutionary search process.

suboptimal zones. As the evolution proceeds, this fraction increases to nearly 46% by iteration 1000, reflecting the growing influence of rule-based chemical filters and adaptive feedback mechanisms. Concurrently, the Pareto front advances steadily, expanding the achievable trade-off frontier and uncovering materials that balance multiple performance criteria. This improvement stems from LLEMA's evolutionary refinement process, where chemically guided mutations, crossover between promising candidates, and memory-based selection pressure iteratively prune the search space. The result is a guided transition from memorized priors to data-driven innovation, leading to denser clusters of thermodynamically stable, property-aligned materials and a measurable improvement in both diversity and discovery quality across generations.

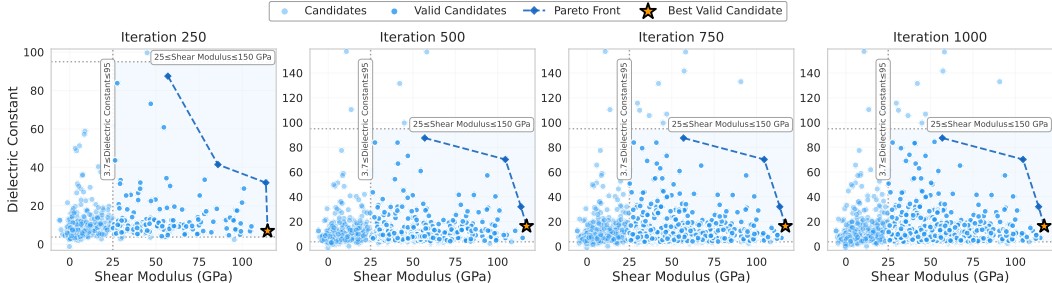

**Figure 9:** Evolution of the Pareto front during multi-objective optimization for SAW/BAW Acoustics substrates.

**Increasing Validity and Structural Fidelity.** As memorization declines, the fraction of valid and physically plausible candidates rises steadily. Early populations contain roughly 30% valid materials, many of which are simple substitutions of known prototypes, while later generations surpass 80% validity. This improvement results from the combined effect of oracle-guided scoring and constraint-aware feedback that prunes chemically inconsistent structures while reinforcing successful design patterns. The process simultaneously expands the search's coverage of the periodic table: early iterations are dominated by light elements and common oxides, whereas later generations incorporate diverse transition metals, alkaline earths, and rare-earth substitutions, reflecting a richer exploration of the underlying chemical landscape.

### E.2 ROBUSTNESS TO PROMPT VARIATIONS

LLM-based generation can be sensitive to how prompts are phrased, so we assess whether LLEMA remains stable under changes in prompt structure. Starting from the base LLEMA prompt, we manually created three alternative versions by rephrasing instructions, reorganizing constraint descriptions, and varying the emphasis on key design objectives. These variants preserve the same semantics while altering the surface form of the prompt. We then evaluate whether LLEMA continues to produce consistent and chemically plausible candidates across these prompt variations. To assess robustness, we performed a focused study on two representative tasks: Wide-Bandgap Semiconductors and Hard, Stiff Ceramics. For each task, we executed the design iterations while varying the prompt formulations used during generation. Across these controlled experiments, we observed only modest variation in the number of valid candidates produced, indicating that the method is reasonably stable under moderate prompt modifications. Results are summarized in Table 7.

**Table 7:** Effect of different prompt formulations on valid candidate generation (hit-rate).

| Prompt Version | Wide-bandgap Semiconductors | Hard, stiff Ceramics |
|---|---|---|
| Version 1 | 24.0 | 41.3 |
| Version 2 | 22.8 | 42.8 |
| Version 3 | 26.8 | 44.5 |

### E.3 EFFECT OF MODEL SCALE AND DOMAIN KNOWLEDGE

Our primary experiments use `GPT-4o-mini` and `Mistral-Small-3.2`, both of which are mid-scale models. We further evaluate open-sourced models like `Qwen2.5-7B/14B/32B` (Yang et al., 2024a) to function within the pipeline. However, generating valid CIF files and proposing plausible materials is not a simple rule-based task, as LLMs must encode enough materials-science domain knowledge to respect crystallographic constraints, chemical compatibility, and physically realistic structures. This explains why performance improves from 7B to 14B/32B with the model size primarily acting as a proxy for embedded scientific priors (see Table 8). This also underscores a central point: *rule-based evolution alone is not enough.* Without domain knowledge, search trajectories drift toward non-physical or chemically invalid structures even if the output format is correct. Larger models succeed not because of scale, but because they carry a richer implicit understanding of stability, bonding, and symmetry. Thus, the framework leverages the domain knowledge encoded in the LLM, not the rules alone.

**Table 8:** Hit rates (%) across tasks for different model sizes.

| Task | Qwen2.5-7B | Qwen2.5-14B | Qwen2.5-32B |
|---|---|---|---|
| High-$k$ Dielectrics | 4.46 | 10.20 | 13.27 |
| SAW/BAW Acoustic Substrates | 1.30 | 20.83 | 23.81 |
| Wide-Bandgap Semiconductors | 9.80 | 22.55 | 20.00 |

### E.4 RELIABILITY OF SURROGATE MODELS

Density Functional Theory (DFT) remains the gold standard for evaluating stability and functional properties; however, its high computational cost (tens of minutes to hours per structure) makes it impractical to incorporate directly into an evolutionary loop. To maintain fast iteration, LLEMA employs surrogate models as lightweight proxies, enabling rapid property estimation while guiding the search toward promising regions of the design space. To assess how well surrogate-guided discoveries translate to high-fidelity physics, we sample 150 valid candidates across four tasks: Wide-Bandgap Semiconductors, Photovoltaic Absorbers, Piezo Energy Harvesters, and Acousto-optic Hybrids, and evaluate them using Quantum ESPRESSO[10] (Giannozzi et al., 2009) using default parameters. Among these candidates, **144 out of 150 (96%) satisfy the task constraints under DFT**, indicating strong agreement between surrogate predictions and high-fidelity outcomes. As shown in Table 9, the fraction of LLEMA-generated candidate materials whose predicted properties are

---

[10]https://github.com/QEF

physically consistent with DFT reference calculations is provided. High DFT validity across all five evaluated properties indicates that the surrogate models used within LLEMA reliably approximate true quantum-mechanical behavior, supporting the robustness of property predictions across diverse materials design tasks.

**Table 9:** DFT Validity of predicted material properties across different tasks.

| Property | DFT Validity |
|---|---|
| band_gap | 97% |
| formation_energy | 99% |
| energy_above_hull | 96% |
| piezo_max_dij | 97% |
| piezo_max_dielectric | 98% |

## E.5 ROBUSTNESS TO NOISE

Since LLEMA 's evolutionary loop depends on surrogate model predictions to guide the search, it is important to assess whether small perturbations in these predictions alter the trajectory of the algorithm. To test robustness, we constructed noisy surrogate models by injecting additive Gaussian noise ($\mu = 0$, $\sigma = 0.05$) into every predicted property at every evaluation step. The rest of the pipeline, including mutation, selection, and validity checks, was kept identical to the standard LLEMA setup. We then re-ran LLEMA with these noisy surrogates across all tasks and compared the resulting candidate sets with those obtained using the original models. While the overall validity rate decreases, LLEMA consistently identifies chemically similar regions of the design space: we observe 100% elemental overlap for Wide-Bandgap Semiconductors and High-k Dielectrics, 96% for SAW/BAW Acoustics, and 84% for Hard, Stiff Ceramics. This high overlap indicates that LLEMA does not simply follow surrogate feedback, but leverages its domain knowledge and evolutionary reasoning to steer the search toward plausible solutions even when the guidance signal is perturbed. To further assess high-fidelity correctness under noisy guidance, we performed DFT calculations on a set of 150 candidates generated with noisy surrogates. **141 out of 150 (94%) satisfy the task constraints under DFT, closely matching the noiseless evaluation.** Together, these results show that LLEMA maintains coherent, physically meaningful trajectories even under uncertain predictions, reinforcing the robustness and reliability of the method.

## E.6 EFFICIENCY ANALYSIS

To evaluate computational efficiency, we compare LLEMA with LLMatDesign and a simple base LLM baseline using prompt token lengths, number of API calls, and per-iteration runtime. Table 10 summarizes these results. Although LLEMA requires roughly $1.5\times$ more tokens per iteration than the base LLM, it achieves an $8–10\times$ improvement in discovery quality, yielding a substantially more favorable performance–compute tradeoff. The base LLM exhibits shorter runtimes primarily because it does not invoke surrogate prediction modules and often defaults to memorized priors or direct database lookups. For LLEMA and LLMatDesign, the reported runtime includes loading pretrained surrogate models and performing property predictions. These overheads can be further reduced via model caching, parallelization across islands, and improved hardware utilization. Overall, LLEMA delivers markedly higher scientific performance while requiring only modest additional compute, making it practical for real-world materials discovery pipelines.

**Table 10:** Efficiency comparison across methods.

| Method | Input Tokens/Call | Output Tokens/Call | Time/Iter. (s) | API Calls/Iter. |
|---|---|---|---|---|
| LLEMA | 563 | 176 | 15.84 | 3 |
| LLMatDesign | 509 | 185 | 13.23 | 3 |
| Base LLM | 457 | 118 | 4.12 | 2 |

## E.7 DIVERSITY OF IDENTIFIED MATERIALS

To assess the chemical diversity introduced by LLEMA, we compared the elemental distributions of generated materials against those proposed by the baseline `GPT-4o-mini` across four representative

discovery tasks: photovoltaic absorbers, hard stiff ceramics, high-$k$ dielectrics, and wide-bandgap semiconductors (Figures 10–13). Each heatmap illustrates the normalized ratio of element occurrences aggregated over 250 LLM-guided iterations. Across all tasks, LLEMA consistently expands the explored chemical space, incorporating a broader range of metallic, semiconducting, and nonmetallic elements compared to the baseline. For instance, in the hard ceramic and high-$k$ dielectric tasks (Figures 11–12), LLEMA identifies richer combinations of transition metals and oxygen-rich compositions—key building blocks for mechanically and electronically robust materials. Similarly, for wide-bandgap semiconductors (Figure 13), LLEMA diversifies the generated candidates beyond conventional group III–V and II–VI chemistries. These results highlight how the integration of memory-based refinement and chemistry-informed evolutionary rules enables LLEMA to navigate and exploit underexplored regions of chemical space, thereby enhancing both diversity and relevance in LLM-driven materials discovery.

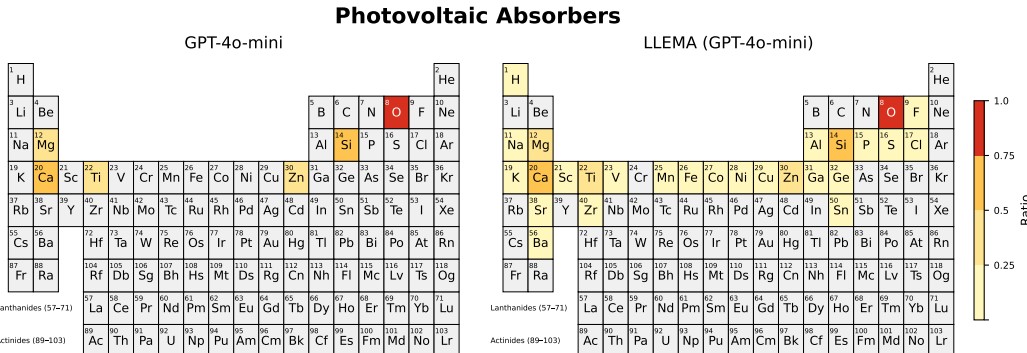

**Figure 10:** Elemental distributions of predicted photovoltaic absorbers after 250 iterations for `GPT-4o-mini` and LLEMA (`GPT-4o-mini`). The heatmap represents the normalized ratio of element occurrence in identified absorber materials.

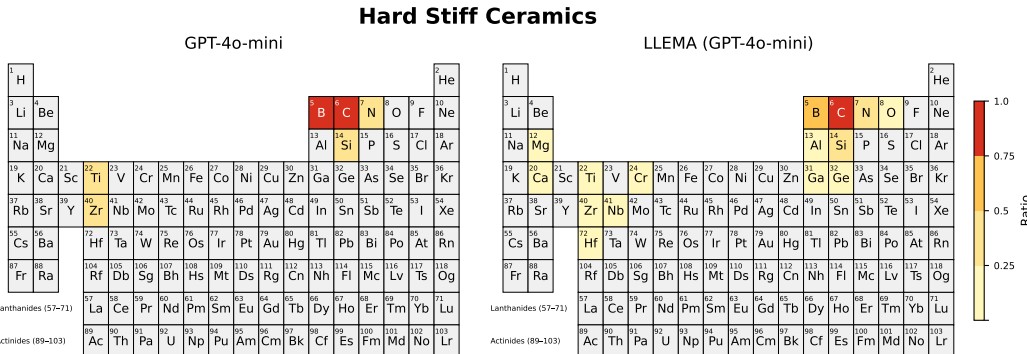

**Figure 11:** Elemental distributions of predicted hard stiff ceramics after 250 iterations for `GPT-4o-mini` and LLEMA (`GPT-4o-mini`). The heatmap shows the normalized ratio of element occurrence in identified ceramic materials.

## High-k dielectric

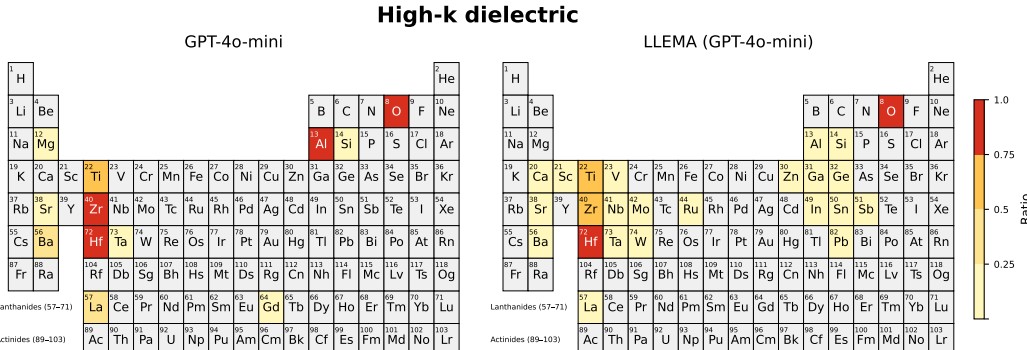

**Figure 12:** Elemental distributions of predicted high-$k$ dielectrics after 250 iterations for `GPT-4o-mini` and LLEMA (`GPT-4o-mini`). The heatmap shows the normalized ratio of element occurrence in identified dielectrics.

## Wide-Bandgap Semiconductors

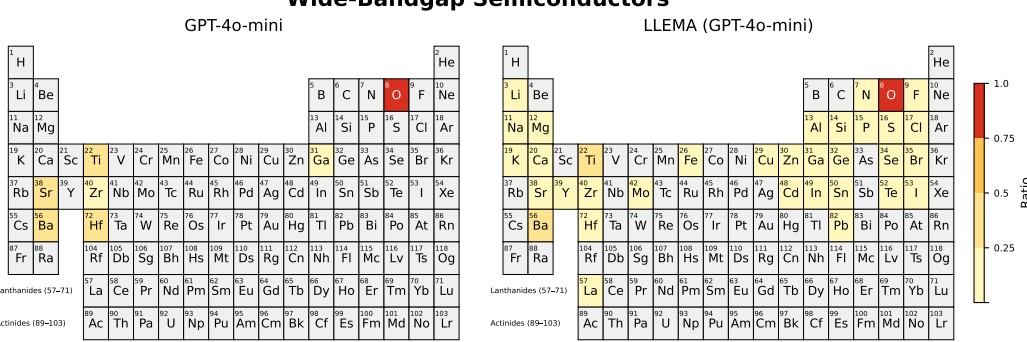

**Figure 13:** Elemental distributions of predicted wide-bandgap semiconductors after 250 iterations for `GPT-4o-mini` and LLEMA (`GPT-4o-mini`). The heatmap shows the normalized ratio of element occurrence in identified semiconductors.

