# OpenReview forum: "LLEMA: Evolutionary Search with LLMs for Multi-Objective Materials Discovery"
_ICLR.cc/2026/Conference — ICLR 2026 Poster_

### Official Review · Reviewer_Hxv1 · 2025-10-20

**Soundness:** 2
**Presentation:** 3
**Contribution:** 2
**Rating:** 4
**Confidence:** 3

**Summary:**

The paper introduce a unified framework called LLEMA to intergrates LLM-knowledge and domain knowledge to generate high-quality sample through iterative search and refinement. The authors test the framwork on 10 diverse tasks and achieve significant performance gain on some of the tasks.

**Strengths:**

1. The framework provides good insight of combining LLM as search engine and modules that provides domain knowledge feedback.
2. Significant improvements are made on some tasks.

**Weaknesses:**

1. The originality of the method is somewhat limited as its high-level framework is similar to LLMatDesign except for some module design like stage b and multi-objective optimization. For example, the implementation of surrogate predictor comes from previous work but provides major improvements according to figure 7.
2. The experimental results are somewhat suspicious. LLEMA achieves very good results on tasks which LLMatDesign has nearly trivial values and has only comparable and slightly worse results on the tasks which LLMatDesign has non-trivial values. Although it's unreasonable to require SOTA result on every task, but more analysis should be made to explain this observation. For example, the author could demonstrate that the performance gain from trivial values to large values is not caused by some invalid setup of baseline methods but the design of some modules through ablation study.
3. Different prompts are used to in candidate generation stage. It is neccessary to clarify how prompt engineering affects the results to illustrates the robustness of the method.

**Questions:**

Please address my concerns in the weakness part. Also please check the accessibility of the code link in the abstract. I'm willing to raise the score if they're properly addressed.

---

> ### Author Response · Authors · 2025-11-22
> **Response to Reviewer Hxv1 (1/2)**
>
> We thank the reviewer for their time and thoughtful feedback, which helped us strengthen the manuscript.
>
> > Comparison with LLMatDesign.
>
> We list the differences below along with their empirical impact on material discovery:
>
> | Aspect                          | **LLEMA**                                                                                                                  | **LLMatDesign**                                                                                                            | **Δ Metrics (vs. LLEMA)**                                   |
> | ------------------------------- | -------------------------------------------------------------------------------------------------------------------------- | --------------------------------------------------------------------------------------------------------------------------- | ------------------------------------------------------------ |
> | **Problem Formulation**         | Realistic multi-objective optimization                                                                                     | Treats discovery as a single-objective task                                                                                | --                                     |
> | **Search Strategy**             | Uses a **population-based, multi-island evolutionary framework**                                                           | Follows a **single linear feedback loop**                                   | HR: 30.2 → 12.9 (–17.3)
> | **Feedback & Refinement**       | Maintains **success and failure memory pools** to guide evolution                                               | **No Memory.** Refers to only the previous candidate                                              | HR: 30.2 → 4.4 (–25.8), Stab: 27.6 → 1.8 (–25.8), Mem: 16.6 → 95.3 (+78.7) |
> | **Property Prediction**         | Integrates SOTA surrogate predictors with an oracle                                                             | Trains a neural network for each property                                                                                  | HR: 30.2 → 4.8 (–25.4), Stab: 27.6 → 3.4 (–24.2)            |
> | **Chemistry-Aware Evolution**   | Uses chemically-informed transformations, ensuring stability and synthesizability                                          | Uses generic transformations                                                                           | HR: 30.2 → 29.8 (–0.4), Stab: 27.6 → 21.5 (–6.1), Mem: 16.6 → 25.3 (+8.7) |
>
> 1. **Why LLMatDesign performs well only on certain tasks.**
>
> LLMatDesign struggles on most tasks for three structural reasons.
> - **Reduces discovery to single-property optimization**, so it cannot handle multi-constraint trade-offs and collapses into unstable regions when several objectives must be satisfied together.
> - **Lacks memory**, operating as a single trajectory that cannot accumulate successes or avoid repeated failures, which limits exploration.
> - Its **generic GA edits are not chemistry-aware**, so generated structures violate basic physical or chemical constraints, leading to invalid or unstable candidates.
>
> The empirical effects of the above shortcomings can also be seen via the table above. Finally, **the contribution of this work includes not only the framework but also the benchmark**, which shifts from theoretical single-property optimization to realistic, multi-objective materials discovery.
>
> 2. **Validating that the gains are due to design choices, not incorrect implementation.**
>
> Each of LLEMA's components contribute to its performance gain. As shown in Section 4, removing core components from LLEMA yields a monotonic degradation. As LLMatDesign lacks any of these components, it is expected to have a low score. Thus, large improvement on multi-objective tasks is driven by LLEMA’s design rather than an invalid baseline.
>
> 3. **Setup consistency.**
>
> All methods operate under the same search budget, iteration count, surrogate model, and crystallographic representation format. This ensures that performance differences arise from algorithmic behavior rather than external factors.
>
> **Design Updates**:
>
> To ensure a fair comparison, we incorporated several upgrades into the LLMatDesign framework so it could operate effectively on our benchmark:
> - We updated LLMatDesign's framework to support multi-objective design.
> - We added LLEMA's crystallographic representation stage (Figure 1, Stage B) into the LLMatDesign framework.
> - Next, we integrated surrogate models into LLMatDesign so that both methods used the exact same evaluation and out-of-distribution samples are considered in the search.
> - Additionally, both LLEMA and LLMatDesign are run using identical search budgets and iteration counts.
>
> These upgrades improve LLMatDesign over its original version, ensuring that our comparison reflects method quality and not implementation gaps.

---

> ### Author Response · Authors · 2025-11-22
> **Response to Reviewer Hxv1 (2/2)**
>
> > Different prompts are used to in candidate generation stage. It is necessary to clarify how prompt engineering affects the results to illustrates the robustness of the method.
>
> Thank you for this important comment. Due to the limited time available during the rebuttal period, we conducted a focused experiment on two representative tasks (Wide-Bandgap Semiconductors and Hard, Stiff Ceramics), while varying the prompt formulations used in the generation stage using the gpt-4o-mini model. Within this constrained study, we observed only modest variation in the number of valid candidates produced, suggesting that the method is reasonably robust to moderate prompt changes. A summary of these results is provided below:
>
> | Prompt Version | Wide-Bandgap Semiconductors | Hard, Stiff Ceramics     |
> |----------------|-----------------------------|--------------------------|
> | Version 1      | 24.0                        |  41.3                    |
> | Version 2      | 22.8                        |  42.8                    |
> | Version 3      | 26.8                        |  44.5                    |
>
> We have updated Appendix D.5 to include the robustness analysis for varying prompts.
>
> > Also please check the accessibility of the code link in the abstract.
>
> Thank you for pointing this out. As the surrogate models such as CGCNN and ALIGNN each maintain their own official GitHub repositories, we do not include them in our repository. Instead, we recommend cloning those repositories directly and placing them inside the `surrogate_models` directory. We have updated the main README and added a dedicated README within the `surrogate_models` folder to make the loading of surrogate models. The updated documentation now provides step-by-step instructions for setting up these models and running the entire workflow.
>
> We hope these updates fully address the accessibility concerns.

---

> > ### Author Response · Authors · 2025-11-28
> > **Looking forward to discussion**
> >
> > Dear Reviewer Hxv1,
> >
> > We hope our answers and new experiments have addressed your concerns and questions. Please let us know if you have any more questions before the end of the discussion period.
> >
> > Should there be no additional concerns, we kindly ask you to consider revising your score.
> >
> > Thank you for your time and thoughtful feedback!

---

> > > ### Author Response · Authors · 2025-11-30
> > > **Reminder from Authors**
> > >
> > > Dear Reviewer Hxv1,
> > >
> > > Thank you for your feedback during the review process! We believe that our detailed response has addressed your concerns. If you have any concerns or questions, please do not hesitate to let us know before the author discussion period ends. We will be happy to answer them during the discussion.
> > >
> > > Should there be no further concerns, we kindly ask that you consider revising your score.
> > >
> > > Thank you,
> > >
> > > Authors

---

### Official Review · Reviewer_zEFw · 2025-10-26

**Soundness:** 3
**Presentation:** 3
**Contribution:** 3
**Rating:** 6
**Confidence:** 3

**Summary:**

This paper presents LLEMA, which employs chemistry-informed evolutionary rules and memory-based refinement to guide large language models (LLM) in multi-objective materials discovery. Its performance is evaluated on 10 different tasks, showing higher hit rates and chemical plausibility (characterized by thermodynamic stability) than baselines including generative models and vanilla LLMs. The importance of components of the proposed method is assessed via ablation tests.

**Strengths:**

- The proposed method combines LLM’s advantage of utilizing unstructured data sources with principled chemical knowledge guidance.
- The ablation studies are comprehensive, in particular, the analysis of memorization vs guided exploration investigates a key question that LLM4Sci should answer.

**Weaknesses:**

- The application scenarios are oversimplifications of materials design. For example, thermodynamic stability is the minimum requirement of chemical plausibility.
- Some common methods for constrained multi-objective optimization are not included in the benchmark tests (see Q1).

**Questions:**

## Technical
1. Would statistically more principled design optimization methods, such as Bayesian optimization, work for the problem setting? How would they compare to the proposed method and other LLM-based or generative methods?
2. In Table 2, the hit rate of generative models such as CDVAE are surprisingly low. What might be the reason? Is the tested task too different from what the methods are designed for?
3. Following up on Q2, more recent generative models, such as MatterGen, are not covered in the benchmark. Would they perform better than the old ones?

## Clarity
4. In Sec. 2.4, how are the islands determined? This should be briefly explained in the main text. Besides, how much would the method choice or randomness in island partitioning affect the method’s performance?
5. Minor issues
    - in Line 33, “materials discovery remains a formidable challenge…” is an overstatement.
    - Line 183, what format does the LLM output, CIF or JSON?

---

> ### Author Response · Authors · 2025-11-22
> **Response to Reviewer zEFw (1/2)**
>
> We thank the reviewer for their time and thoughtful feedback, which helped us strengthen the manuscript.
>
> > The application scenarios are oversimplifications of materials design. For example, thermodynamic stability is the minimum requirement of chemical plausibility.
>
> We thank the reviewer for the thoughtful comment. We agree that thermodynamic stability is a key prerequisite for chemical plausibility. While full kinetic synthesizability across multiple microstructural scales is beyond our current scope, **prior work has shown meaningful connections between thermodynamic stability and synthesizability** [1]. Using hierarchical evaluation of materials at multiple length and time scales, or direct experimental integration would be significantly more expensive and time-consuming, and are an active area of research [2]. Thus, overall, thermodynamic stability remains a practical and well-supported proxy for chemical plausibility for accelerating material discovery.
>
> > Some common methods for constrained multi-objective optimization are not included such as Bayesian optimization. How would they compare to the proposed method and other LLM-based or generative methods?
>
> We thank the reviewer for this insightful comment. While methods like **Bayesian Optimization (BO)** can be used for materials design, they **face a qualitative limitation** since the models underlying these tools (e.g. Gaussian Process in Bayesian optimization) have **to ‘re-learn’ fundamental correlations from scratch using only limited experimental data**. These methods assume a low-dimensional continuous design space with a fixed parameterization of candidates. In materials discovery, the space is discrete and combinatorial, and there is no continuous embedding that allows BO to directly propose valid CIFs or satisfy property constraints. We believe **LLMs** are better suited to overcome this limitation as they **can start the search owing to their vast domain knowledge of material science**. For example, when searching for metallic alloys, an LLM could immediately suggest valid candidates like the Pt-Au alloy family. In contrast, **BO, without an expert-curated search space and initial dataset, would be forced to re-learn basic concepts** like the superior properties of Pt-Au alloys over pure components, thus struggling to find novel materials efficiently.
>
> > In Table 2, the hit rate of generative models such as CDVAE are surprisingly low. What might be the reason? Is the tested task too different from what the methods are designed for?
>
> We thank the reviewer for this observation. Generative baselines such as CDVAE, G-SchNet, and DiffCSP exhibit low hit rates primarily because they **do not integrate natural language or task-specific contextual information** into their generation process and function as unconditional generators. They rely purely on stochastic sampling from a distribution and generate new candidates without considering property constraints, or material context. Consequently, the generation process is random rather than goal-driven, and there is **no mechanism for feedback or iterative refinement based on property evaluation**. As a result, generative methods perform poorly in comparison with LLM-based methods. This further showcases the novelty of our benchmark that tests material discovery in real world scenarios.
>
> > Following up on Q2, more recent generative models, such as MatterGen, are not covered in the benchmark. Would they perform better than the old ones?
>
> We thank the reviewer for raising this point. As shown in Table below, MatterGen achieves substantially higher validity rates than the generative baselines, but it still underperforms significantly relative to LLEMA. This gap is expected given the fundamental differences between the approaches as discussed in the earlier response.
>
> | Task                               | MatterGen | LLEMA |
> | ---------------------------------- | --------- |---- |
> | Hard, Stiff Ceramics               | 8.2  | **42.9** |
> | Photovoltaic Absorbers             | 2.3  | **22.9** |
> | Solid-State Electrolytes           | 5.3  | **46.2** |
> | Stable Wide-Bandgap Semiconductors | 6.5  | **21.6** |
> | SAW/BAW Acoustic Substrates        | 15.6 | **25.9** |
> | Structural Materials for Aerospace | 0.0  | **1.0**  |
> | High-k Dielectrics                 | 0.6  | **19.9** |
> | Hard Coating Materials             | 2.1  | **17.8** |
> | Acousto-optic Hybrids              | 11.2 | **22.9** |
>
> We have added MatterGen as one of the baselines in table 2 of section 3.3 and thank the reviewer for raising the question.
>
> ---
>
> References:
>
> [1] Aykol, Muratahan, et al. "Thermodynamic limit for synthesis of metastable inorganic materials." Science advances 4.4 (2018).
>
> [2] Abolhasani, Milad, and Eugenia Kumacheva. "The rise of self-driving labs in chemical and materials sciences." Nature Synthesis 2.6 (2023).

---

> ### Author Response · Authors · 2025-11-22
> **Response to Reviewer zEFw (2/2)**
>
> > In Sec. 2.4, how are the islands determined? This should be briefly explained in the main text. Besides, how much would the method choice or randomness in island partitioning affect the method’s performance?
>
> The islands are formed by simply splitting the population into *k* subpopulations. The main effect of choosing *k* is how the total iteration budget is divided. For example:
>
> - 1,000 total iterations - 10 islands → ~100 iterations per island
> - 1,000 total iterations - 1 island → 1,000 iterations for that island
>
> As each island evolves independently, **the number of islands balances between exploration (more islands) and exploitation (fewer islands)**. Our results reflect this expected behavior, where we represent two extremes: (i) 1 island and (ii) 10 islands and choosing a *k*=5 which operates between these two extremes. Our results below are as expected where **moderate island counts** (e.g., *k* = 5) **tends to balance the trade-off**, while too many islands (*k* = 10) reduce the refinement depth.
>
> | # Islands | Wide-Bandgap Semicond. | High-k Dielectrics | SAW/BAW Acoustics | Hard, Stiff Ceramics |
> |----------|-------------------------|--------------------|-------------------|----------------------|
> | 1        |  18.62         |  5.59          |  22.00         |  5.65         |
> | 5        |  33.62         |  19.96         |  59.88         |  30.99         |
> | 10       |  24.46         |  20.34         |  26.78         |  16.19         |
>
> The island count simply allocates the iteration budget across parallel search paths. As long as *k* is chosen to match the total budget (e.g., more islands when the budget is large), the method behaves reliably and consistently across domains.
>
> We have added a short subsection in the implementation details (Appendix C.1) to explain this.
>
> > in Line 33, “materials discovery remains a formidable challenge…” is an overstatement.
>
> We appreciate the reviewer’s comment, but we respectfully maintain that describing materials discovery as a "formidable challenge" is accurate.
>
> **Search Space**: Even restricting to compounds to 4 elements per compound, the theoretical design space exceeds 10¹² quaternary compositions, and feasible materials occupy only a tiny, highly fragmented subset. It would take thousands of years even under generous assumptions regarding discovery to just explore this space of 4 elements per compound. Multi-objective applications in industry-specific tasks further reduce the viable region. Given this combination of enormous search space and extremely narrow feasible regions, we believe that characterizing materials discovery as a formidable challenge is justified.
>
>
> **Policy Evidence**: The White House’s 2011 **Materials Genome Initiative (MGI)**[1] identifies advanced materials as essential to economic security and human well-being, **emphasizing the difficulty and urgency of accelerating materials discovery**. This perspective is reinforced by the **Department of Energy’s National Strategy for Advanced Manufacturing**[2] and the **Department of Defense’s National Defense Science & Technology Strategy**[3], both of which **designate advanced materials as critical to national security and technological leadership**. Our wording is not an overstatement but a reflection of the widely recognized importance and complexity of accelerating materials discovery.
>
> > Line 183, what format does the LLM output, CIF or JSON?
>
> Thank you for pointing it out, we will make the change to CIF in the final version.
>
> ---
>
> **References:**
>
> [1] Materials Genome Initiative for Global Competitiveness, 2011.
>
> [2] U.S. Department of Energy, National Strategy for Advanced Manufacturing, 2022.
>
> [3] U.S. Department of Defense, National Defense Science & Technology Strategy, 2023.

---

> > ### Author Response · Authors · 2025-11-28
> > **Looking forward to discussion**
> >
> > Dear Reviewer zEFw,
> >
> > We hope our answers and new experiments have addressed your concerns and questions. Please let us know if you have any more questions before the end of the discussion period.
> >
> > Should there be no additional concerns, we kindly ask you to consider revising your score.
> >
> > Thank you for your time and thoughtful feedback!

---

> > ### Comment · Reviewer_zEFw · 2025-11-28
> >
> > I appreciate the Authors' efforts in improving the paper. My comments regarding clarity are well addressed. However, some issues remain:
> > 1. Baseline. Compared to BO and generative models (GenAI), LLM-guided design seems to be aiming for a balance between efficiency (GenAI's edge) and validity (BO's edge). Yet, LLEMA is compared to GenAI in validity; BO is ignored in the benchmark. I partially agree with the Authors' statement that BO faces challenges in combinatorial space, but there have been works addressing that, e.g., methods [[1]](https://arxiv.org/abs/2011.02004) [[2]](https://arxiv.org/abs/1806.08838); applications [[3]](https://arxiv.org/abs/2412.17283).
> > 2. The oversimplification of materials design tasks. It's OK to use thermodynamic stability as a *proxy* of chemical plausibility, but claiming the materials satisfying this minimal requirement as *chemically plausible* is not rigorous.
> > 3. The "policy evidence" shows the importance of materials discovery, not that it's formidable. I think the wording is OK in more specific terms, e.g., materials discovery in combinatorial space, but in general, materials discovery have observed continuous progress, so saying it's "formidable" is an overclaim.
> >
> > In summary, this is a good paper with limitations (mainly point 1), and I view my current rating as fair.

---

> ### Author Response · Authors · 2025-11-30
> **Author Response to Reviewer zEFw**
>
> > Bayesian Optimization (BO) Baseline
>
> We thank the reviewer for highlighting relevant Bayesian Optimization (BO) based works. They have greatly helped us identify a valuable baseline for comparison. Based on your recommendations, we have incorporated a BO baseline into our pipeline using the BOCS formulation[1], where each evolutionary choice is encoded as a binary vector and evaluated through our full materials-generation loop. The results are summarized below:
>
> | Task  | BO (19 Rules) | BO-Base (4 Simple Rules) | LLEMA (GPT) |
> |-----|--------|---------|-------|
> | Stable Wide-Bandgap Semiconductors | 1.16% | 2.23%  | 21.62%   |
> | Hard–Stiff Ceramics                | 6.32% | 0.70%  | 42.94%   |
> | High-k Dielectrics                 | 5.23% | 8.05%  | 19.96%   |
>
> BO struggles in this setting because it assumes a smooth objective and learns only from scalar validity scores, whereas the materials space is discrete and highly non-smooth. This leads BO to prefer simple rules (add, remove, substitute, etc.) with small changes that maintain CIF validity, thus exploring only a narrow region of the search space.
>
> In contrast, the LLM-driven optimization loop can use the full set of transformations because it leverages domain knowledge, reasoning traces, and memory to generate and refine candidates even when the underlying search landscape is highly non-smooth. This enables LLEMA to explore chemically richer trajectories that BO’s surrogate never learns to utilize. As a result, the observed gap in hit rate reflects BO’s tendency to overfit to low-variance, low-diversity rule choices, whereas the LLM-centered process can exploit broader transformations that BO systematically avoids.
>
>
> > The oversimplification of materials design tasks - thermodynamic stability as a proxy of chemical plausibility
>
> We thank the reviewer for the clarification and fully agree that thermodynamic stability is only a minimal proxy for chemical plausibility. In our work, it only serves as an initial feasibility filter. Moreover, our stability evaluation is modular and can be expanded with more comprehensive criteria as needed. Extending the framework to full synthesis or kinetic realism is valuable future work, but beyond the current scope.
>
> > The "policy evidence" shows the importance of materials discovery, not that it's formidable. I think the wording is OK in more specific terms, e.g., materials discovery in combinatorial space, but in general, materials discovery has observed continuous progress, so saying it's "formidable" is an overclaim.
>
> We appreciate the reviewer’s point. To avoid any misunderstanding, we will adjust the wording in line 33 to make this scope explicit, for example, "However, materials discovery remains challenging due to the immense combinatorial space...."
> This maintains accuracy while addressing the reviewer’s concern.
>
> [1] Baptista, Ricardo, and Matthias Poloczek. "Bayesian optimization of combinatorial structures." In International conference on machine learning, pp. 462-471. PMLR, 2018.
>
> **Thank you**: We sincerely appreciate the reviewer’s thoughtful questions, which allowed us to strengthen the manuscript and provide a thorough comparison. If the reviewer feels that all concerns have now been fully addressed, we kindly request that they consider updating the evaluation score accordingly.

---

### Official Review · Reviewer_BxZr · 2025-11-01

**Soundness:** 2
**Presentation:** 3
**Contribution:** 2
**Rating:** 4
**Confidence:** 3

**Summary:**

The paper proposes an LLM-guided evolutionary framework for generating candidate materials under explicit structural and property constraints. In each iteration, an LLM is prompted to produce crystallographically specified or composition-level candidates; chemistry-informed evolutionary rules and a success/failure memory then filter and refine these candidates; finally, a surrogate (learned) oracle estimates multiple target properties and a multi-objective scorer selects promising samples to feed back into the next round.

**Strengths:**

1. The proposed method puts the LLM inside an iterative optimization loop (LLM $\rightarrow$ screen $\rightarrow$ refine $\rightarrow$ re-prompt). That aligns with current best practices in LLM-for-science/LLM-as-optimizer work and makes the contribution intelligible from an ML perspective.

2. The framework explicitly tries to stay inside chemically plausible regions via rule-based evolution and feasibility checks, rather than treating materials generation as free-text generation. That’s an important realism step for this domain.

3. The paper shows improvements on several tasks and on Pareto fronts (not just single-property top-1), making the method look more generally useful.

**Weaknesses:**

1. Several recent works also use pretrained LLMs to propose materials/structures and then improve them using an external scorer (e.g., [1]). The paper needs to spell out what is actually new here. Right now, the novelty can look like “a solid engineering combination” rather than a clearly new algorithmic design.

2. The paper claims to “leverage scientific knowledge embedded in LLMs,” but it does not show: (1) performance when replacing the LLM with a lighter template-based (or rule-based) generator; (2) performance across LLM scales or domain specialization; or (3) how often the LLM actually proposes candidates that could not be reached by rules alone. Without that, it’s hard to justify the LLM cost.

3. Surrogate reliability in out-of-distribution regions is not fully addressed. Evolutionary loops guided by learned oracles can be misled by over-optimistic predictions. The paper mentions memory-based refinement, but does not clearly report surrogate error over iterations or how many top candidates were re-evaluated with higher-fidelity methods. This weakens the “accelerating materials discovery” claim.

4. Evaluation realism is partly unclear. Many of the reported constraints (stability, synthesizability, plausibility) can be enforced by heuristics or low-cost predictors. It is not obvious how many of the “hits” would remain if checked with higher-fidelity simulation or expert curation.

5. The framework seems to rely on hand-crafted chemistry-informed rules. It is not clear how much effort is needed to move from the materials considered here to, say, MOFs, polymers, or battery electrolytes.

**Questions:**

1. Can you provide quantitative comparisons to the most similar recent frameworks that also do “LLM proposal + evolutionary/active search” for materials (e.g., [1])? Right now, the difference to those systems is mostly described at a high level.


2. What happens if you (1) use a smaller/general-purpose model; (2) use a materials-tuned LLM; or (3) replace the LLM with a rule/template generator? This is important to justify that the gain comes from LLM knowledge, not only from the evolutionary loop.


3. The current version seems to rely solely on surrogate-augmented oracles for property evaluation. I did not find evidence of a higher-fidelity (e.g., DFT) re-evaluation of the top candidates. Could you provide such validation or at least report surrogate error for late-stage candidates?


4. Since every generation invokes an LLM under constraints, what is the total token/runtime cost across the 14 tasks? Are there caching or retrieval-augmentation tricks to keep the loop affordable?

[1] Gan J, Zhong P, Du Y, et al. Large language models are innate crystal structure generators[C]//AI for Accelerated Materials Design-ICLR 2025. 2025.

---

> ### Author Response · Authors · 2025-11-22
> **Response to Reviewer BxZr (1/4)**
>
> We thank the reviewer for their time and thoughtful feedback, which helped us strengthen the manuscript.
>
> > Several recent works also use pretrained LLMs to propose materials/structures and then improve them using an external scorer (e.g., [1]). The paper needs to spell out what is actually new here, as, the novelty can look like “a solid engineering combination” rather than a clearly new algorithmic design.
>
> We thank the reviewer for this question. LLEMA is the first LLM-based framework that tackles real-world materials discovery. The **novelty lies in this task formulation of benchmark and the algorithmic components necessary to make it solvable.**
>
> **Real-World Materials Discovery**: Prior methods reduce discovery to stable crystal generation [1] or single-property optimization [2], ignoring the multi-constraint nature of real world discovery. Our benchmark reflects this by jointly considering functional, chemical and stability requirements. LLEMA addresses this real world setting, unlike unconditioned generators that do not support task-specific discovery.
>
> **Problem Complexity**: Perfectly stable material can be completely unusable if it fails the functional criteria, for example, discovering high-k dielectrics, requires candidates to *simultaneously* satisfy dielectric response, band gap along with stability. This combination creates a narrow, application-specific feasible region. Thus, previous works focus on crystal-stability[1] optimization which at best is a random crystal generator that does not address discovery of any specific material.
>
> **LLEMA Design**:  Our ablations (Section 4) show the tangible impact of each component when integrated in LLEMA. The table below summarizes how LLEMA’s step-by-step integration affects performance for each challenge, highlighting the contribution of each incremental component.
>
> | **Challenge**  | **Component Added**                       | **Δ Hit Rate / Δ Stability** |
> |--------|--------|--------|
> | **Task Conditioning** | LLM | +4.4   / +1.8             |
> | **Memorization** | + Memory                         | +10.7 / +18.3             |
> | **Efficient exploration** | + Multi-island evolution        | +14.7 / +1.7              |
> | **Chemically plausible evolution**  | + Chemistry-informed rules   | +0.4  / +5.8              |
>
> - **Task Conditioning**: Unguided sampling in the huge chemical and structural space leads to irrelevant candidates.  LLEMA conditions the LLM on the task specification and feedback, enabling targeted exploration using the LLM’s domain reasoning rather than random search.
>
> - **Memorization**: When iterated naïvely, LLMs generate repetitive structures, limiting to a small memorized region of search space. To showcase LLEMA's ability to explore diverse functional groups, we have added a dedicated section in Appendix D.2, with Figures 11-14 highlighting the diversity across four tasks.
>
> - **Efficient Exploration**: Materials search spaces often contain disjoint regions, and a single evolutionary trajectory can easily collapse. LLEMA’s multi-island framework mitigates this by running independent exploratory trajectories in parallel, enabling better exploration.
>
> - **Chemically plausible evolution**: Real world materials discovery requires candidates to be synthesizable and meet chemical principles. Generic GA operators leads to unstable materials. LLEMA incorporates chemistry-informed evolution operators directly into the generative step to ensure proposals remain chemically viable.
>
> In summary, LLEMA is an algorithmic framework with design choices directly driven by the nature of materials discovery.
>
> > Can you provide quantitative comparisons to the most similar recent frameworks that also do “LLM proposal + evolutionary/active search” for materials (e.g., [1])?
>
> Prior work [1] does not perform task-conditioned, multi-objective discovery; it operates as an unconditional “LLM + evolution’’ generator focused on structural refinement and stability similar to baselines CDVAE, DiffCSP, G-SchNet and MatterGen [3]. The closest "LLM proposal + evolutionary/active search" related method is [2] (refer Table 2) , which targets single-property optimization and suffers from limited exploration and memorization.
>
> We have updated Table 2 in our manuscript to include [3].
>
> ---
>
> **References:**
>
> [1] Gan, J. et al. “MatLLMSearch: Crystal Structure Discovery with Evolution-Guided Large Language Models.” arXiv preprint arXiv:2502.20933 (2025).
>
> [2] Jia, S. et al. “LLMatDesign: Autonomous Materials Discovery with Large Language Models.” arXiv preprint arXiv:2406.13163 (2024).
>
> [3] Zeni, C. et al. “A generative model for inorganic materials design.” Nature 2025.

---

> ### Author Response · Authors · 2025-11-22
> **Response to Reviewer BxZr (2/4)**
>
> > Evaluation realism is partly unclear. Many of the reported constraints (stability, synthesizability, plausibility) can be enforced by heuristics or low-cost predictors. It is not obvious how many of the “hits” would remain if checked with higher-fidelity simulation or expert curation.
>
> Thank you for the constructive feedback. Our response is in two parts:
>
> - **High-fidelity calculations**: We agree that DFT calculations on the successful candidates could confirm stability. Due to the limited time during the rebuttal period and the time-consuming nature DFT calculations, we could perform DFT validation only on a subset of the valid candidates. We performed DFT evaluations using Quantum ESPRESSO[1] with default settings. As DFT remains costly (≈30 minutes–16 hours per property), we evaluated a representative subset of 150 valid candidates, uniformly sampled from four tasks: Wide-Bandgap Semiconductors, Photovoltaic Absorbers, Piezoelectric Harvesters, and Acousto-optic Hybrids using the official Quantum ESPRESSO GitHub workflow. We report **144 out of 150 successful samples (96%) following the constraints**, indicating **strong consistency between the surrogate-model predictions and DFT outcomes**. We further provide a property specific breakdown of DFT evaluations.
>
> | Property              | Valid |
> |-----------------------|-------|
> | band_gap              |   97%    |
> | formation_energy      |   99%    |
> | energy_above_hull     |   96%    |
> | piezo_max_dij         |    97%   |
> | piezo_max_dielectric  |    98%   |
>
> - **Expert evaluation of final candidates**: Importantly, the **final discovered compounds are reviewed by expert materials scientists** for plausibility and  (Section 3.4, Discovered Candidates). For instance, several of the proposed compositions, such as **CaZnSi and MgZnSi oxides do not appear in existing literature or databases**, indicating that they are not duplicates or minor variants of known phases. At the same time, these candidates remain chemically related to established ZnO-based families, demonstrating that LLEMA explores novel but plausible regions of chemical space.
>
> > Surrogate reliability in out-of-distribution regions is not fully addressed.
>
> We agree with the reviewer’s comment on the reliability of surrogate models. However, these surrogate models are state-of-the-art in the material science community[2], and are selected based on continually updated benchmarks[3], making them the **most practical alternative to costly DFT evaluations.** Existing discovery frameworks face similar limitations from approximate simulation tools[4] or restricted experimental throughput[5]. Yet such frameworks routinely uncover novel candidates in large design spaces[6].
>
> To address the reliability concerns of surrogate models, we added Gaussian noise (mean =0; standard dev =0.05) to the outputs of the surrogate models. While, the overall validity drops, we observe that 100% elemental overlap for Wide-Bandgap Semiconductors and High-k Dielectrics; 96% overlap for SAW/BAW Acoustics and 84% for Hard, Stiff Ceramics. Additionally, **141 out of 150 (94%) of the valid candidates satisfy the task constraints under DFT**, closely matching the noiseless evaluation.
>
> This shows that exploration does not suffer under noisy feedback from surrogate models, owing to the stabilizing effect of its domain-aware LLM proposals, memory buffers, and chemistry-informed evolutionary rules. Given the high computational cost of DFT evaluations, this analysis further *reinforces the reliability of the surrogate-guided optimization process in accelerating materials discovery*. We sincerely appreciate the reviewer’s suggestion, which has strengthened the our design choice.
>
> We have updated Appendix D.4 in the manuscript with the above experiments.
>
> ---
>
> **References**:
>
> [1] Giannozzi, Paolo, et al. "QUANTUM ESPRESSO: a modular and open-source software project for quantumsimulations of materials." Journal of physics: Condensed matter 21.39 (2009): 395502.
>
> [2] Dunn, Alexander, et al. "Benchmarking materials property prediction methods: the Matbench test set and Automatminer reference algorithm." npj Computational Materials 6.1 (2020): 138.
>
> [3] Riebesell, Janosh, et al. "Matbench Discovery--A framework to evaluate machine learning crystal stability predictions." arXiv preprint arXiv:2308.14920 (2023).
>
> [4] Herbol, Henry C., Matthias Poloczek, and Paulette Clancy. "Cost-effective materials discovery: Bayesian optimization across multiple information sources." Materials Horizons 7.8 (2020): 2113-2123
>
> [5] Kusne, A. Gilad, et al. "On-the-fly closed-loop materials discovery via Bayesian active learning." Nature communications 11.1 (2020): 5966.
>
> [6] Arróyave, Raymundo, et al. "A perspective on Bayesian methods applied to materials discovery and design." MRS communications 12.6 (2022): 1037-1049.

---

> ### Author Response · Authors · 2025-11-22
> **Response to Reviewer BxZr (3/4)**
>
> > The framework seems to rely on hand-crafted chemistry-informed rules. It is not clear how much effort is needed to move from the materials considered here to, say, MOFs, polymers, or battery electrolytes.
>
> We thank the reviewer for this insightful comment. Our goal in this work is to **demonstrate a general and largely automated framework for materials discovery**. The chemistry-informed rules we employ are grounded in basic, universally accepted principles simply to prevent unphysical generations, not to encode domain-specific knowledge. These are **not extensive hand-crafted heuristics but rather the fundamental constraints a materials scientist would apply when proposing plausible candidates**. As MOF properties are strongly dependent on crystal structure, LLEMA can be directly applied for MOFs, and slight variations would enable its use for other classes such as battery electrolytes or polymers. Importantly, these adaptations are standard in materials discovery, far smaller than the highly tailored frameworks commonly used today. Developing generalizable, cross-domain discovery frameworks is an active area of research, and LLEMA is a strong step toward achieving such broadly applicable automation.
>
> > Since every generation invokes an LLM under constraints, what is the total token/runtime cost across the 14 tasks? Are there caching or retrieval-augmentation tricks to keep the loop affordable?
>
> | Method        | Input Tokens/Call | Output Tokens/Call | Total Time/iteration (s) | API calls/iteration |
> |:---:|:---:|:---:| :---:|:---:|
> | LLEMA         | 563     | 176      | 15.84      | 3  |
>
> Since we report the results in main table for 1000 iterations, the average run time of LLEMA is ~5 hours with average token count include input and input 2,200,000 tokens. We do not believe that caching would be productive since each LLM call ideally would query information for a new chemical compound. However, retrieval-augmentation tricks could be explored under the assumption that a high quality database exists but remains out of the scope of the current paper since we do not have access to a high quality offline database. Regardless, we believe that our current framework is extremely affordable given it takes ~1.75 USD per task for 1000 iterations.

---

> ### Author Response · Authors · 2025-11-22
> **Response to Reviewer BxZr (4/4)**
>
> > The paper claims to “leverage scientific knowledge embedded in LLMs” but it does not show:
> > (1) performance when replacing the LLM with a lighter template-based generator;
>
> Thank you for the thoughtful question.
>
> A template-based generator collapses back to the behavior of unconditional generative models as it enumerates candidates without task specific grounding. This has two main limitations:
>
> (i) Low chemical plausibility: Template-driven outputs often violate simple stability heuristics or produce non-realistic structures. In contrast, LLMs provide (a) chemically informed priors, and (b) the ability to translate between natural-language constraints and structured representations (CIF), which is essential for property prediction.
>
> (ii) Lack of feedback: Template based generators cannot learn from previous iterations and hence sample uniformly from a large space of possible combinations. It is estimated that 10^12 quaternary compounds exist [1]. Assuming it takes 1 minute for property predictions, it would end up taking 2 million years to just explore quaternary compounds. LLMs significantly cut down this time where our framework yields multiple promising candidates in under 10 hours of runtime.
>
> > (2) performance across LLM scales or domain specialization;
>
> 1. Our primary experiments use GPT-4o-mini and Mistral-24B both of which are mid-scale models. We further evaluate using Qwen2.5-7B/14B/32B.
>
> | Task   | Qwen2.5-7B     | Qwen2.5-14B     | Qwen2.5-32B     |
> |-------|-----|------|----|
> | High-k Dielectrics | 4.46%  | 10.20%  | 13.27%  |
> | SAW/BAW Acoustic Substrates  | 1.30%  | 20.83%  | 23.81%  |
> | Stable Wide-Bandgap Semiconductors  | 9.80%  | 22.55%  | 21.50%  |
>
> Importantly, this highlights that simple rule-based evolution is not sufficient. Generating valid CIF files and proposing plausible materials is not a simple rule-based task as LLMs must encode enough materials-science domain knowledge to respect crystallographic constraints, chemical compatibility, and physically realistic structures. Larger models perform better because they bring richer implicit understanding of stability, bonding, and symmetry, enabling more meaningful exploration. Thus, the framework leverages the domain knowledge encoded in the LLM, not the rules alone.
>
> 2. Domain-adapted LLMs such as [2] would further strengthen performance. These models typically encode priors about stability trends, compositional feasibility, and structure-property relationships. In our framework, such priors directly improve search efficiency because the generator proposes candidates that are better aligned with thermodynamic and task-specific constraints. However in interest of generalization and ease to use, we only use general purpose models.
>
> > (3) how often the LLM actually proposes candidates that could not be reached by rules alone.
>
> While an LLM alone tends to produce repetitive candidates (over 90% recalled from Materials Project), its strength emerges when it is integrated into LLEMA’s evolutionary loop. The LLM can interpret feedback, apply chemistry-informed rules, and interact with memory buffers, showcasing capabilities that a template-based generator cannot support. To clarify the contribution of each component in LLEMA, we include the table (from Section 4) below, which compares plain LLM generation, mutation/crossover, and our chemistry-informed rules:
>
> | Method                      | H.R↑ | Stab.↑ | Mem.↓ |
> |-----------------------------|------|--------|-------|
> | **LLEMA**                   | **30.2** | **27.6** | **16.6** |
> | *w/o* rules                 | 29.8 | 21.5   | 25.3  |
> | Base LLM                         | 4.4  | 1.8    | 95.3  |
>
> ---
>
> **References:**
>
> [1] Davies, Daniel W. et al. "Computational Screening of All Stoichiometric Inorganic Materials", Chem, Volume 1, Issue 4, 617 - 627
>
> [2] Gruver, Nate, et al. "Fine-tuned language models generate stable inorganic materials as text." arXiv preprint arXiv:2402.04379 (2024).

---

> > ### Author Response · Authors · 2025-11-28
> > **Looking forward to discussion**
> >
> > Dear Reviewer BxZr,
> >
> > We hope our answers and new experiments have addressed your concerns and questions. Please let us know if you have any more questions before the end of the discussion period.
> >
> > Should there be no additional concerns, we kindly ask you to consider revising your score.
> >
> > Thank you for your time and thoughtful feedback!

---

> > > ### Author Response · Authors · 2025-11-30
> > > **Reminder from Authors**
> > >
> > > Dear Reviewer BxZr,
> > >
> > > Thank you for your feedback during the review process! We believe that our detailed response has addressed your concerns. If you have any concerns or questions, please do not hesitate to let us know before the author discussion period ends. We will be happy to answer them during the discussion.
> > >
> > > Should there be no further concerns, we kindly ask that you consider revising your score.
> > >
> > > Thank you,
> > >
> > > Authors

---

### Official Review · Reviewer_EVF3 · 2025-11-03

**Soundness:** 3
**Presentation:** 3
**Contribution:** 3
**Rating:** 6
**Confidence:** 4

**Summary:**

The proposed framework LLEMA is designed for efficient materials discovery, which includes an LLM-driven candidate generator with chemistry-aware evolutionary rules, memory pools of successful or failed candidates, and ML-based property predicting oracles. The authors designed ten materials design tasks with multi-constraint targets and stability screening. The effectiveness of the framework is demonstrated through the high valid hit rates and Pareto fronts than both generative models and LLM-only baselines. The key contribution of the work is its demonstration of materials discovery through constraining the LLM’s search with domain rules and iterative feedback, while grounding candidates with a surrogate oracle and enforcing stability.

**Strengths:**

1. The work is a good demonstration of employing LLMs while injecting chemistry heuristics for 'multi-objective and synthesizability-aware materials design'.
2. The diverse benchmarking tasks are well designed which align with realistic multi-constraint targets

**Weaknesses:**

1. No DFT validation provided on the top novel successful candidates for higher-fidelity stability evaluation.
2. The quality of the accepted or rejected designs of the pool is highly dependent on the reliability of surrogate models. Exploration is potentially limited due to the unintentional error cumulated, which the system would favor correct known designs.
3. The efficiency of the proposed framework is not fully explained.

**Questions:**

1. It is mentioned in A.2 that equal weights are assigned to each property in multi-objective scoring, but the section also suggests prioritized weighting. What are the specific weighting strategy used in each of the tasks?
2. In the same section, it is also stated that  'performance-critical properties' prioritized over 'feasibility constraints' like formation energy, density, and hull stability. Is this the case for all tasks? Can you justify the design? An unstable crystal is still invalid with optimal shear modulus. How likely will the design sacrifice stability for certain performance metrics?
3. How does the distribution of formation energy evolve over iterations in the tasks, e.g. stable wide-bandgap semiconductor task?
4. Have you examined the failure pool for failure modes? The oracles can mislabel novel candidates as failures due to OOD, e.g. novel composition missing from the patched phase diagram or absent from the Materials Project API. Will this error being cumulated over iterations and lead to a conservative search bias toward known chemistries?
5. How do you measure novelty and diversity beyond Materials Project, to ensure that the generated designs are not near-duplicates of known phases?
6. What are the average prompt token lengths, number of LLM calls, API calls and pool sizes when executing each task? How does LLEMA compare to baselines in terms of efficiency and resources required, e.g. inference time and cost.

---

> ### Author Response · Authors · 2025-11-22
> **Response to Reviewer EVF3 (1/3)**
>
> We thank the reviewer for their time and thoughtful feedback, which helped us strengthen the manuscript.
>
> > No DFT validation provided on the top novel successful candidates for higher-fidelity stability evaluation.
>
> Thank you for the constructive feedback, we agree that DFT calculations on the successful candidates could confirm stability. Due to the limited time during the rebuttal period and the time-consuming nature DFT calculations, we could perform DFT validation only on a subset of the valid candidates. Specifically, we used the Quantum ESPRESSO[1] GitHub repository and we evenly sampled a total of 150 valid candidates from four tasks: Wide-Bandgap Semiconductors, Photovoltaic Absorbers, Piezo-electric Harvesters, and Acousto-optic Hybrids. We report **144 out of 150 successful samples (96%) following the constraints**, indicating **strong consistency between the surrogate-model predictions and DFT outcomes**. We further provide a property specific breakdown of DFT evaluations.
>
> | Property              | Valid |
> |-----------------------|-------|
> | band_gap              |   97%    |
> | formation_energy      |   99%    |
> | energy_above_hull     |   96%    |
> | piezo_max_dij         |    97%   |
> | piezo_max_dielectric  |    98%   |
>
> Given the high computational cost of DFT evaluations, this analysis further *reinforces the reliability of the surrogate-guided optimization process in accelerating materials discovery*. We sincerely appreciate the reviewer’s suggestion, which has strengthened our results.
>
> We have updated the manuscript with the above results which can be found at Appendix D.3.
>
> > The quality of the accepted or rejected designs of the pool is highly dependent on the reliability of surrogate models. Exploration is potentially limited due to the unintentional error cumulated, which the system would favor correct known designs.
>
> We agree with the reviewer’s comment on the reliability of surrogate models. However, these surrogate models are state-of-the-art in the material science community[2], and are selected based on continually updated benchmarks[3], making them the **most practical alternative to costly DFT evaluations** (often >16 hours per structure for some tasks). Existing discovery frameworks face similar limitations from approximate simulation tools[4] or restricted experimental throughput[5]. Yet such frameworks routinely uncover novel candidates[6].
>
> However, to evaluate the effect of noise on LLEMA we added Gaussian noise (mean =0; standard dev =0.05) to the outputs of the surrogate models. While, the overall validity drops, we observe that **100% elemental overlap for Wide-Bandgap Semiconductors and High-k Dielectrics; 96% overlap for SAW/BAW Acoustics and 84% for Hard, Stiff Ceramics**. Additionally, **141 out of 150 (94\%) of the valid candidates satisfy the task constraints under DFT**, closely matching the noiseless evaluation. This shows that **exploration does not suffer under noisy feedback** from surrogate models, owing to the stabilizing effect of its domain-aware LLM proposals, memory buffers, and chemistry-informed evolutionary rules.
>
> We have updated Appendix D.4 in the manuscript with the above noise sensitivity experiment.
>
> ---
>
> **References:**
>
> [1] Giannozzi, Paolo, et al. "QUANTUM ESPRESSO: a modular and open-source software project for quantumsimulations of materials." Journal of physics: Condensed matter 21.39 (2009): 395502.
>
> [2] Dunn, Alexander, et al. "Benchmarking materials property prediction methods: the Matbench test set and Automatminer reference algorithm." npj Computational Materials 6.1 (2020): 138.
>
> [3] Riebesell, Janosh, et al. "Matbench Discovery--A framework to evaluate machine learning crystal stability predictions." arXiv preprint arXiv:2308.14920 (2023).
>
> [4] Herbol, Henry C., Matthias Poloczek, and Paulette Clancy. "Cost-effective materials discovery: Bayesian optimization across multiple information sources." Materials Horizons 7.8 (2020): 2113-2123
>
> [5] Kusne, A. Gilad, et al. "On-the-fly closed-loop materials discovery via Bayesian active learning." Nature communications 11.1 (2020): 5966.
>
> [6] Arróyave, Raymundo, et al. "A perspective on Bayesian methods applied to materials discovery and design." MRS communications 12.6 (2022): 1037-1049.

---

> ### Author Response · Authors · 2025-11-22
> **Response to Reviewer EVF3 (2/3)**
>
> > It is mentioned in A.2 that equal weights are assigned to each property in multi-objective scoring, but the section also suggests prioritized weighting. What are the specific weighting strategy used in each of the tasks?
>
> We thank the reviewer for pointing this out. In naïve multi-objective formulation, we consider aggregating all the objectives using a sum function with a weight assigned for each objective. The weight of the i-th objective can indeed be considered a hyperparameter, and determining these weights is often nontrivial[1]. To avoid introducing this additional layer of complexity, we assign equal weights to all primary objectives, for example, band gap and formation energy in the Wide-Bandgap Semiconductor task. After calculating the primary objective scores, we evaluate thermodynamic stability analysis, which is not part of the weighted objective.
>
> We have updated Appendix A.2 and slightly modified the scoring function in Section 2.4 to prevent any misunderstanding.
>
> > In the same section, it is also stated that 'performance-critical properties' prioritized over 'feasibility constraints' like formation energy, density, and hull stability. Is this the case for all tasks? Can you justify the design? An unstable crystal is still invalid with optimal shear modulus. How likely will the design sacrifice stability for certain performance metrics?
>
> We thank the reviewer for the question. To clarify: in all tasks, stability is a strict feasibility requirement, not an tunable objective. The multi-objective score aggregates only the performance properties explicitly defined in the task constraints, while stability is evaluated afterward as an independent feasibility check. As stated in Section 3.2, Hit-Rate counts only candidates that satisfy all property constraints, and Stability measures the fraction of those that are thermodynamically stable. Thus, a material may meet all performance constraints yet still be rejected as unstable. Our earlier phrasing (“performance-critical properties prioritized over feasibility constraints”) referred solely to the ordering of the scoring pipeline, not to any relaxation of stability requirements.
> We have revised the Appendix section A.2 for clarification.
>
> > How does the distribution of formation energy evolve over iterations in the tasks, e.g. stable wide-bandgap semiconductor task?
>
> Across tasks including the stable wide-bandgap semiconductor task, the per-island plots show a clear trend: early iterations contain a wide range of formation energies, while later iterations concentrate around lower values. This reflects the effect of iterative selection and refinement, where unstable candidates are filtered out and each island progressively moves toward more favorable regions of the search space.
> We have added Appendix section D.6 to include the plots for the above.
>
> > Have you examined the failure pool for failure modes? The oracles can mislabel novel candidates as failures due to OOD, e.g. novel composition missing from the patched phase diagram or absent from the Materials Project API. Will this error being cumulated over iterations and lead to a conservative search bias toward known chemistries?
>
> We appreciate the reviewer highlighting this important issue. We agree that relying solely on oracles could mislabel novel candidates, and **we explicitly designed LLEMA to avoid such the "OOD missing data -> failure" pattern**.
>
> **Oracle+Surrogate Design**: In LLEMA, *candidates are not marked as failures simply because they are absent from the oracle*. Our property module is hierarchical: we query the oracle first that stores DFT values, and when data are missing we fall back to pretrained surrogate models to estimate properties for out-of-distribution compositions and structures. **A candidate enters the failure pool only when its predicted properties violate task constraints and not because it is novel (or absent from the Materials Project API).** Because missing data candidates are routed through surrogates rather than labeled as failures, the system *does not accumulate systematic penalties for novelty.* This breaks the positive feedback loop that would otherwise bias evolution toward known regions.
>
> Empirically, Section 4.3 shows that **restricting to the oracle alone collapses hit rates (<5%) and biases the search toward known chemistries**, whereas including surrogate models restores exploration. Section 4.1 further shows that exact oracle matches decrease under LLEMA, indicating exploration beyond familiar phases.
>
> ---
>
> **References**
>
> [1] Nathanael Kusanda, Gary Tom, Riley Hickman, AkshatKumar Nigam, Kjell Jorner, and Alan AspuruGuzik. Assessing multi-objective optimization of molecules with genetic algorithms against relevant baselines.

---

> ### Author Response · Authors · 2025-11-22
> **Response to Reviewer EVF3 (3/3)**
>
> > How do you measure novelty and diversity beyond Materials Project, to ensure that the generated designs are not near-duplicates of known phases?
>
> To comprehensively evaluate the diversity and novelty of LLEMA's output beyond the initial overlap analysis presented in Section 4.1, **we engaged material scientists to evaluate the candidate structures**, as detailed in our Discovered Materials analysis (Section 3.4). For instance, several of the proposed compositions, such as CaZnSi and MgZnSi oxides for photovoltaics tasks do not appear in existing literature or databases, indicating that they are not duplicates or minor variants of known phases. At the same time, these candidates remain chemically related to established ZnO-based families, demonstrating that LLEMA explores novel but plausible regions of chemical space.
>
> As for diversity, we plot out the elemental distributions of the suggested materials and find out that LLEMA is significantly more diverse than base LLM suggestions. We have updated the manuscript appendix section D.2 to address the above.
>
> > The efficiency of the proposed framework is not fully explained.
> > What are the average prompt token lengths, number of LLM calls, API calls and pool sizes when executing each task? How does LLEMA compare to baselines in terms of efficiency and resources required, e.g. inference time and cost.
>
> | Method        | Input Tokens/Call | Output Tokens/Call | Total Time/iteration (s) | API calls/iteration |
> |:---:|:---:|:---:| :---:|:---:|
> | LLEMA         | 563     | 176      | 15.84      | 3              |
> | LLMatDesign   | 509     | 185      | 13.23      | 3            |
> | Base LLM      | 457     | 118      | 4.12       | 2              |
>
> Despite using ~1.5× more tokens per iteration, LLEMA achieves an 8–10× improvement over the base LLM, demonstrating a substantially better performance–compute tradeoff. The shorter runtime of the base LLM is largely because of not invoking any surrogate prediction modules. As the base LLM often defaults to memorized knowledge or simple API lookups, the resulting low execution times are lesser as compared to the other two baselines. For LLEMA and LLMatDesign, execution time includes loading the pretrained model and performing inference, which can be further reduced through standard optimizations such as model caching and improved resource utilization. Overall, LLEMA delivers dramatically higher performance for only modest additional compute.
>
> We have updated Appendix F to include the above details.

---

> > ### Author Response · Authors · 2025-11-28
> > **Looking forward to discussion**
> >
> > Dear Reviewer EVF3,
> >
> > We hope our answers and new experiments have addressed your concerns and questions. Please let us know if you have any more questions before the end of the discussion period.
> >
> > Should there be no additional concerns, we kindly ask you to consider revising your score.
> >
> > Thank you for your time and thoughtful feedback!

---

> > > ### Author Response · Authors · 2025-11-30
> > > **Reminder from Authors**
> > >
> > > Dear Reviewer EVF3,
> > >
> > > Thank you for your feedback during the review process! We believe that our detailed response has addressed your concerns. If you have any concerns or questions, please do not hesitate to let us know before the author discussion period ends. We will be happy to answer them during the discussion.
> > >
> > > Should there be no further concerns, we kindly ask that you consider revising your score.
> > >
> > > Thank you,
> > >
> > > Authors

---

### Author Response · Authors · 2025-11-22
**General Response to Reviewers**

We thank all reviewers for their thoughtful feedback. All reviewers consistently highlight the strength of our framework, noting the effective integration of LLM guidance with chemistry-aware rules and feasibility checks (EVF3, BxZr, zEFw, Hxv1) and the realistic multi-constraint task suite (EVF3). They also recognize the novelty of our iterative LLM-in-the-loop approach and the combination of proposal, rule-constrained evolution, and memory-guided refinement (BxZr, zEFw). Reviewers further emphasize our strong empirical performance, citing high valid hit rates, strong Pareto fronts, and improvements across several tasks (EVF3, BxZr, zEFw, Hxv1). Finally, they value the depth of our analysis, including comprehensive ablations and the memorization-vs-guided-exploration study (zEFw, Hxv1). additionally, the constructive feedback provided by the reviewers has been invaluable in improving the paper. We have provided detailed responses to each reviewer's comments individually and outline the key results from new experiments conducted in response to multiple reviewer requests.

**New Experiments**

In the updated manuscript, all clarifications and additions are marked in blue text, and sections that were frequently referenced during the reviews are highlighted in red text to make them easier to locate. A few figures and tables have been adjusted for better readability, without altering any underlying data or results.

- **DFT validation** (EVF3, BxZr): We added a DFT evaluation of a representative subset of valid candidates using Quantum ESPRESSO to confirm surrogate-model consistency (Appendix D.3).

- **Noise robustness of surrogates** (EVF3, BxZr): We added an experiment injecting controlled noise into surrogate predictions to test whether exploration degrades under imperfect feedback (Appendix D.4).

- **Clarifications on weighting and stability** (EVF3): We clarified the multi-objective weighting scheme and the role of stability as a strict feasibility filter (Appendix A.2).

- **Formation-energy evolution** (EVF3): Added per-island plots showing how stability metrics evolve over iterations (Appendix D.6).

- **Failure-pool, novelty, and diversity analysis** (EVF3, BxZr): Expanded explanations of oracle–surrogate interactions and added novelty/diversity visualizations and expert-review examples (Appendix D.2 & Section 3.4).

- **Efficiency analysis** (EVF3, Hxv1): Added average token usage, API calls, and per-iteration runtime comparisons with baselines (Appendix F), along with a case study illustrating how LLEMA improves through the evolution process (Appendix D.1).

- **Multi-island ablation** (zEFw, Hxv1): Added an ablation in Appendix A.2 varying the number of islands to show its effect on exploration and refinement.

- **Additional baselines and LLM-scale analysis** (zEFw, BxZr): Incorporated MatterGen into the benchmark (Table 2) and added results using multiple Qwen model scales (Appendix E).

We are happy to answer any further questions/concerns.

---

### Meta-Review · Area_Chair_DeaV · 2026-01-06

**Summary:**

This paper introduces a framework that includes an LLM and a set of chemistry-informed evolutionary rules for multi-objective materials discovery. The authors show higher objective values than competing approaches and demonstrate that the proposed materials are viable, supported by DFT validation. While the method proposed in the paper is very similar to existing LLM-based materials discovery pipelines and thus incremental from a novelty perspective, its performance is noteworthy and will be of interest to the materials discovery community. The authors have conducted careful science, including resource requirements, detailed experimental design descriptions, and robustness studies. While this paper would benefit from a more careful comparison against Bayesian optimization methods, using proper state-of-the-art approaches and going beyond the quick rebuttal experiment, it is nevertheless a contribution that will be of interest to the community, and therefore, I recommend acceptance.

**Reviewer Concerns:**

The concerns that were resolved from reviewer discussion include (1) DFT validation of proposed materials, (2) details about efficiency/resource requirements, (3) questions about the experimental design, and (4) robustness to prompt engineering.

The concerns that were not resolved include (1) the novelty of the method, especially in comparison to LLMatDesign, and (2) comparison to BO benchmarks. While these discussions may not have been resolved even after reviewer discussion, I believe that what the paper lacks in methodological novelty it makes up for in its performance. The slight differences between the existing methods yield strong performance improvements. As stated above, while a more robust BO comparison would benefit the paper, it stands without one.

**Reviewer Scores:**

* Reviewers EVF3, BxZr, and Hxv1 did not engage at all during the discussion period, and so would likely not have changed their scores if they had had the full discussion period.
* Reviewer zEFw felt that the discussion resolved their concerns and maintained their positive score. Given that all points were resolved, it is unlikely that further discussion would have changed their score.

---

### Decision · Program_Chairs · 2026-01-26

Accept (Poster)